# Generalization Properties of Learning with Random Features

**Alessandro Rudi** *
INRIA - Sierra Project-team,
École Normale Supérieure, Paris,
75012 Paris, France
alessandro.rudi@inria.fr

**Lorenzo Rosasco**
University of Genova,
Istituto Italiano di Tecnologia,
Massachusetts Institute of Technology.
lrosasco@mit.edu

## Abstract

We study the generalization properties of ridge regression with random features in the statistical learning framework. We show for the first time that $O(1/\sqrt{n})$ learning bounds can be achieved with only $O(\sqrt{n}\log n)$ random features rather than $O(n)$ as suggested by previous results. Further, we prove faster learning rates and show that they might require more random features, unless they are sampled according to a possibly problem dependent distribution. Our results shed light on the statistical computational trade-offs in large scale kernelized learning, showing the potential effectiveness of random features in reducing the computational complexity while keeping optimal generalization properties.

## 1 Introduction

Supervised learning is a basic machine learning problem where the goal is estimating a function from random noisy samples [1, 2]. The function to be learned is fixed, but unknown, and flexible non-parametric models are needed for good results. A general class of models is based on functions of the form,

$$f(x) = \sum_{i=1}^{M} \alpha_i \, q(x, \omega_i), \tag{1}$$

where $q$ is a non-linear function, $\omega_1, \ldots, \omega_M \in \mathbb{R}^d$ are often called centers, $\alpha_1, \ldots, \alpha_M \in \mathbb{R}$ are coefficients, and $M = M_n$ could/should *grow* with the number of data points $n$. Algorithmically, the problem reduces to computing from data the parameters $\omega_1, \ldots, \omega_M, \alpha_1, \ldots, \alpha_M$ and $M$. Among others, one-hidden layer networks [3], or RBF networks [4], are examples of classical approaches considering these models. Here, parameters are computed by considering a non-convex optimization problem, typically hard to solve and analyze [5]. Kernel methods are another notable example of an approach [6] using functions of the form (1). In this case, $q$ is assumed to be a positive definite function [7] and it is shown that choosing the centers to be the input points, hence $M = n$, suffices for optimal statistical results [8, 9, 10]. As a by product, kernel methods require only finding the coefficients $(\alpha_i)_i$, typically by convex optimization. While theoretically sound and remarkably effective in small and medium size problems, memory requirements make kernel methods unfeasible for large scale problems.

Most popular approaches to tackle these limitations are randomized and include sampling the centers at random, either in a data-dependent or in a data-independent way. Notable examples include Nyström [11, 12] and random features [13] approaches. Given random centers, computations still

reduce to convex optimization with potential big memory gains, provided that the centers are fewer than the data-points. In practice, the choice of the number of centers is based on heuristics or memory constraints, and the question arises of characterizing theoretically which choices provide optimal learning bounds. Answering this question allows to understand the statistical and computational trade-offs in using these randomized approximations. For Nyström methods, partial results in this direction were derived for example in [14] and improved in [15], but only for a simplified setting where the input points are fixed. Results in the statistical learning setting were given in [16] for ridge regression, showing in particular that $O(\sqrt{n} \log n)$ random centers uniformly sampled from $n$ training points suffices to yield $O(1/\sqrt{n})$ learning bounds, *the same as full kernel ridge regression*.

A question motivating our study is whether similar results hold for random features approaches. While several papers consider the properties of random features for approximating the kernel function, see [17] and references therein, fewer results consider their generalization properties.

Several papers considered the properties of random features for approximating the kernel function, see [17] and references therein, an interesting line of research with connections to sketching [24] and non-linear (one-bit) compressed sensing [18]. However, only a few results consider the generalization properties of learning with random features.

An exception is one of the original random features papers, which provides learning bounds for a general class of loss functions [19]. These results show that $O(n)$ random features are needed for $O(1/\sqrt{n})$ learning bounds and choosing less random features leads to worse bounds. In other words, these results suggest that that computational gains come at the expense of learning accuracy. Later results, see e.g. [20, 21, 22], essentially confirm these considerations, albeit the analysis in [22] suggests that fewer random features could suffice if sampled in a problem dependent way.

In this paper, we focus on the least squares loss, considering random features within a ridge regression approach. Our main result shows, under standard assumptions, that the estimator obtained with a number of random features proportional to $O(\sqrt{n} \log n)$ achieves $O(1/\sqrt{n})$ learning error, that is the *same* prediction accuracy of the *exact* kernel ridge regression estimator. In other words, there are problems for which random features can allow to drastically reduce computational costs *without* any loss of prediction accuracy. To the best of our knowledge this is the first result showing that such an effect is possible. Our study improves on previous results by taking advantage of analytic and probabilistic results developed to provide sharp analyses of kernel ridge regression. We further present a second set of more refined results deriving fast convergence rates. We show that indeed fast rates are possible, but, depending on the problem at hand, a larger number of features might be needed. We then discuss how the requirement on the number of random features can be weakened at the expense of typically more complex sampling schemes. Indeed, in this latter case either some knowledge of the data-generating distribution or some potentially data-driven sampling scheme is needed. For this latter case, we borrow and extend ideas from [22, 16] and inspired from the theory of statical leverage scores [23]. Theoretical findings are complemented by numerical simulation validating the bounds.

The rest of the paper is organized as follows. In Section 2, we review relevant results on learning with kernels, least squares and learning with random features. In Section 3, we present and discuss our main results, while proofs are deferred to the appendix. Finally, numerical experiments are presented in Section 4.

## 2   Learning with random features and ridge regression

We begin recalling basics ideas in kernel methods and their approximation via random features.

**Kernel ridge regression**   Consider the supervised problem of learning a function given a training set of $n$ examples $(x_i, y_i)_{i=1}^n$, where $x_i \in X$, $X = \mathbb{R}^D$ and $y_i \in \mathbb{R}$. Kernel methods are nonparametric approaches defined by a *kernel* $K : X \times X \to \mathbb{R}$, that is a symmetric and positive definite (PD) function[2]. A particular instance is kernel ridge regression given by

$$\widehat{f}_\lambda(x) = \sum_{i=1}^n \alpha_i K(x_i, x), \quad \alpha = (\mathbf{K} + \lambda n I)^{-1} y. \tag{2}$$

Here $\lambda > 0$, $y = (y_1, \ldots, y_n)$, $\alpha \in \mathbb{R}^n$, and $\mathbf{K}$ is the $n$ by $n$ matrix with entries $\mathbf{K}_{ij} = K(x_i, x_j)$. The above method is standard and can be derived from an empirical risk minimization perspective [6], and is related to Gaussian processes [3]. While KRR has optimal statistical properties– see later– its applicability to large scale datasets is limited since it requires $O(n^2)$ in space, to store $\mathbf{K}$, and roughly $O(n^3)$ in time, to solve the linear system in (2). Similar requirements are shared by other kernel methods [6].

To explain the basic ideas behind using random features with ridge regression, it is useful to recall the computations needed to solve KRR when the kernel is linear $K(x, x') = x^\top x'$. In this case, Eq. (2) reduces to standard ridge regression and can be equivalenty computed considering,

$$\widehat{f}_\lambda(x) = x^\top \widehat{w}_\lambda \qquad \widehat{w}_\lambda = (\widehat{X}^\top \widehat{X} + \lambda n I)^{-1} \widehat{X}^\top y. \tag{3}$$

where $\widehat{X}$ is the $n$ by $D$ data matrix. In this case, the complexity becomes $O(nD)$ in space, and $O(nD^2 + D^3)$ in time. Beyond the linear case, the above reasoning extends to inner product kernels

$$K(x, x') = \phi_M(x)^\top \phi_M(x') \tag{4}$$

where $\phi_M : X \to \mathbb{R}^M$ is a finite dimensional (feature) map. In this case, KRR can be computed considering (3) with the data matrix $\widehat{X}$ replaced by the $n$ by $M$ matrix $\widehat{S}_M^\top = (\phi(x_1), \ldots, \phi(x_n))$. The complexity is then $O(nM)$ in space, and $O(nM^2 + M^3)$ in time, hence much better than $O(n^2)$ and $O(n^3)$, as soon as $M \ll n$. Considering only kernels of the form (4) can be restrictive. Indeed, classic examples of kernels, e.g. the Gaussian kernel $e^{-\|x - x'\|^2}$, do not satisfy (4) with finite $M$. It is then natural to ask if the above reasoning can still be useful to reduce the computational burden for more complex kernels such as the Gaussian kernel. Random features, that we recall next, show that this is indeed the case.

**Random features with ridge regression**     The basic idea of random features [13] is to relax Eq. (4) assuming it holds only approximately,

$$K(x, x') \approx \phi_M(x)^\top \phi_M(x'). \tag{5}$$

Clearly, if one such approximation exists the approach described in the previous section can still be used. A first question is then for which kernels an approximation of the form (5) can be derived. A simple manipulation of the Gaussian kernel provides one basic example.

**Example 1** (Random Fourier features [13]).  *If we write the Gaussian kernel as $K(x, x') = G(x - x')$, with $G(z) = e^{-\frac{1}{2\sigma^2}\|z\|^2}$, for a $\sigma > 0$, then since the inverse Fourier transform of $G$ is a Gaussian, and using a basic symmetry argument, it is easy to show that*

$$G(x - x') \quad = \quad \frac{1}{2\pi Z} \int \int_0^{2\pi} \sqrt{2}\cos(w^\top x + b) \; \sqrt{2}\cos(w^\top x' + b) \; e^{-\frac{\sigma^2}{2}\|w\|^2} dw \; db$$

*where $Z$ is a normalizing factor. Then, the Gaussian kernel has an approximation of the form (5) with $\phi_M(x) = M^{-1/2}(\sqrt{2}\cos(w_1^\top x + b_1), \ldots, \sqrt{2}\cos(w_M^\top x + b_M))$, and $w_1, \ldots, w_M$ and $b_1, \ldots, b_M$ sampled independently from $\frac{1}{Z}e^{-\sigma^2\|w\|^2/2}$ and uniformly in $[0, 2\pi]$, respectively.*

The above example can be abstracted to a general strategy. Assume the kernel $K$ to have an integral representation,

$$K(x, x') = \int_\Omega \psi(x, \omega)\psi(x', \omega)d\pi(\omega), \quad \forall x, x' \in X, \tag{6}$$

where $(\Omega, \pi)$ is probability space and $\psi : X \times \Omega \to \mathbb{R}$. The random features approach provides an approximation of the form (5) where $\phi_M(x) = M^{-1/2}(\psi(x, \omega_1), \ldots, \psi(x, \omega_M))$, and with $\omega_1, \ldots, \omega_M$ sampled independently with respect to $\pi$. Key to the success of random features is that kernels, to which the above idea apply, abound– see Appendix E for a survey with some details.

**Remark 1** (Random features, sketching and one-bit compressed sensing).  *We note that specific examples of random features can be seen as form of sketching [24]. This latter term typically refers to reducing data dimensionality by random projection, e.g. considering*

$$\psi(x, \omega) = x^\top \omega,$$

*where $\omega \sim N(0, I)$ (or suitable bounded measures). From a random feature perspective, we are defining an approximation of the linear kernel since $\mathbb{E}[\psi(x, \omega)\psi(x', \omega)] = \mathbb{E}[x^\top \omega \omega^\top x'] = x^\top \mathbb{E}[\omega \omega^\top]x' = x^\top x'$. More general non-linear sketching can also be considered. For example in one-bit compressed sensing [18] the following random features are relevant,*

$$\psi(x, \omega) = sign(x^\top \omega)$$

*with $w \sim N(0, I)$ and $sign(a) = 1$ if $a > 0$ and $-1$ otherwise. Deriving the corresponding kernel is more involved and we refer to [25] (see Section E in the appendixes).*

Back to supervised learning, combining random features with ridge regression leads to,

$$\widehat{f}_{\lambda, M}(x) := \phi_M(x)^\top \widehat{w}_{\lambda, M}, \quad \text{with} \quad \widehat{w}_{\lambda, M} := (\widehat{S}_M^\top \widehat{S}_M + \lambda I)^{-1} \widehat{S}_M^\top \widehat{y}, \tag{7}$$

for $\lambda > 0$, $\widehat{S}_M^\top := n^{-1/2} (\phi_M(x_1), \dots, \phi_M(x_n))$ and $\widehat{y} := n^{-1/2} (y_1, \dots, y_n)$.

Then, random features can be used to reduce the computational costs of full kernel ridge regression as soon as $M \ll n$ (see Sec. 2). However, since random features rely on an approximation (5), the question is whether there is a loss of prediction accuracy. This is the question we analyze in the rest of the paper.

## 3   Main Results

In this section, we present our main results characterizing the generalization properties of random features with ridge regression. We begin considering a basic setting and then discuss fast learning rates and the possible benefits of problem dependent sampling schemes.

### 3.1   $O(\sqrt{n} \log n)$ Random features lead to $O(1/\sqrt{n})$ learning error

We consider a standard statistical learning setting. The data $(x_i, y_i)_{i=1}^n$ are sampled identically and independently with respect to a probability $\rho$ on $X \times \mathbb{R}$, with $X$ a separable space (e.g. $X = \mathbb{R}^D$, $D \in \mathbb{N}$). The goal is to minimize the expected risk

$$\mathcal{E}(f) = \int (f(x) - y)^2 d\rho(x, y),$$

since this implies that $f$ will generalize/predict well new data. Since we consider estimators of the form (2), (7) we are potentially restricting the space of possible solutions. Indeed, estimators of this form can be naturally related to the so called reproducing kernel Hilbert space (RKHS) corresponding to the PD kernel $K$. Recall that, the latter is the function space $\mathcal{H}$ defined as as the completion of the linear span of $\{K(x, \cdot) \: : \: x \in X\}$ with respect to the inner product $\langle K(x, \cdot), K(x', \cdot) \rangle := K(x, x')$ [7]. In this view, the best possible solution is $f_{\mathcal{H}}$ solving

$$\min_{f \in \mathcal{H}} \mathcal{E}(f). \tag{8}$$

We will assume throughout that $f_{\mathcal{H}}$ exists. We add one technical remark useful in the following.

**Remark 2.** *Existence of $f_{\mathcal{H}}$ is not ensured, since we consider a potentially infinite dimensional RKHS $\mathcal{H}$, possibly universal [26]. The situation is different if $\mathcal{H}$ is replaced by $\mathcal{H}_R = \{f \in \mathcal{H} \: : \: \|f\| \leq R\}$, with $R$ fixed a priori. In this case a minimizer of risk $\mathcal{E}$ always exists, but $R$ needs to be fixed a priori and $\mathcal{H}_R$ can't be universal. Clearly, assuming $f_{\mathcal{H}}$ to exist, implies it belongs to a ball of radius $R_{\rho, \mathcal{H}}$. However, our results do not require prior knowledge of $R_{\rho, \mathcal{H}}$ and hold uniformly over all finite radii.*

The following is our first result on the learning properties of random features with ridge regression.

**Theorem 1.** *Assume that $K$ is a kernel with an integral representation (6). Assume $\psi$ continuous, such that $|\psi(x, \omega)| \leq \kappa$ almost surely, with $\kappa \in [1, \infty)$ and $|y| \leq b$ almost surely, with $b > 0$. Let $\delta \in (0, 1]$. If $n \geq n_0$ and $\lambda_n = n^{-1/2}$, then a number of random features $M_n$ equal to*

$$M_n = c_0 \ \sqrt{n} \ \log \frac{108\kappa^2 \sqrt{n}}{\delta},$$

*is enough to guarantee, with probability at least $1 - \delta$, that*

$$\mathcal{E}(\widehat{f}_{\lambda_n, M_n}) - \mathcal{E}(f_{\mathcal{H}}) \leq \frac{c_1 \log^2 \frac{18}{\delta}}{\sqrt{n}}.$$

*In particular the constants $c_0, c_1$ do not depend on $n, \lambda, \delta$, and $n_0$ does not depends on $n, \lambda, f_{\mathcal{H}}, \rho$.*

The above result is presented with some simplifications (e.g. the assumption of bounded output) for sake of presentation, while it is proved and presented in full generality in the Appendix. In particular, the values of all the constants are given explicitly. Here, we make a few comments. The learning bound is the same achieved by the *exact* kernel ridge regression estimator (2) choosing $\lambda = n^{-1/2}$, see e.g. [10]. The theorem derives a bound in a worst case situation, where no assumption is made besides existence of $f_{\mathcal{H}}$, and is optimal in a minmax sense [10]. This means that, in this setting, as soon as the number of features is order $\sqrt{n} \log n$, the corresponding ridge regression estimator has optimal generalization properties. This is remarkable considering the corresponding gain from a computational perspective: from roughly $O(n^3)$ and $O(n^2)$ in time and space for kernel ridge regression to $O(n^2)$ and $O(n\sqrt{n})$ for ridge regression with random features (see Section 2). Consider that taking $\delta \propto 1/n^2$ changes only the constants and allows to derive bounds in expectation and almost sure convergence (see Cor. 1 in the appendix, for the result in expectation).

The above result shows that there is a whole set of problems where computational gains are achieved without having to trade-off statistical accuracy. In the next sections we consider what happens under more benign assumptions, which are standard, but also somewhat more technical. We first compare with previous works since the above setting is the one more closely related.

**Comparison with [19].**   This is one of the original random features paper and considers the question of generalization properties. In particular they study the estimator

$$\widehat{f}_R(x) = \phi_M(x)^\top \widehat{\beta}_{R,\infty}, \quad \widehat{\beta}_{R,\infty} = \underset{\|\beta\|_\infty \leq R}{\operatorname{argmin}} \frac{1}{n} \sum_{i=1}^n \ell(\phi_M(x_i)^\top \beta, y_i),$$

for a fixed $R$, a Lipshitz loss function $\ell$, and where $\|w\|_\infty = \max\{|\beta_1|, \cdots, |\beta_M|\}$. The largest space considered in [19] is

$$\mathcal{G}_R = \left\{ \int \psi(\cdot, \omega)\beta(\omega) d\pi(\omega) \;\middle|\; |\beta(\omega)| < R \text{ a.e.} \right\}, \tag{9}$$

rather than a RKHS, where $R$ is fixed a priori. The best possible solution is $f^*_{\mathcal{G}_R}$ solving $\min_{f \in \mathcal{G}_R} \mathcal{E}(f)$, and the main result in [19] provides the bound

$$\mathcal{E}(\widehat{f}_R) - \mathcal{E}(f^*_{\mathcal{G}_R}) \lesssim \frac{R}{\sqrt{n}} + \frac{R}{\sqrt{M}}, \tag{10}$$

This is the first and still one the main results providing a statistical analysis for an estimator based on random features for a wide class of loss functions. There are a few elements of comparison with the result in this paper, but the main one is that to get $O(1/\sqrt{n})$ learning bounds, the above result requires $O(n)$ random features, while a smaller number leads to worse bounds. This shows the main novelty of our analysis. Indeed we prove that, considering the square loss, fewer random features are sufficient, hence allowing computational gains without loss of accuracy. We add a few more tehcnical comments explaining : 1) how the setting we consider covers a wider range of problems, and 2) why the bounds we obtain are sharper. First, note that the functional setting in our paper is more general in the following sense. It is easy to see that considering the RKHS $\mathcal{H}$ is equivalent to consider $\mathcal{H}_2 = \left\{ \int \psi(\cdot, \omega)\beta(\omega)d\pi(\omega) \;\middle|\; \int |\beta(\omega)|^2 d\pi(\omega) < \infty \right\}$ and the following inclusions hold $\mathcal{G}_R \subset \mathcal{G}_\infty \subset \mathcal{H}_2$. Clearly, assuming a minimizer of the expected risk to exists in $\mathcal{H}_2$ *does not* imply it belongs to $\mathcal{G}_\infty$ or $\mathcal{G}_R$, while the converse is true. In this view, our results cover a wider range of problems. Second, note that, this gap is not easy to bridge. Indeed, even if we were to consider $\mathcal{G}_\infty$ in place of $\mathcal{G}_R$, the results in [19] could be used to derive the bound

$$\mathbb{E}\; \mathcal{E}(\widehat{f}_R) - \mathcal{E}(f^*_{\mathcal{G}_\infty}) \lesssim \frac{R}{\sqrt{n}} + \frac{R}{\sqrt{M}} + A(R), \tag{11}$$

where $A(R) := \mathcal{E}(f^*_{\mathcal{G}_R}) - \mathcal{E}(f^*_{\mathcal{G}_\infty})$ and $f^*_{\mathcal{G}_\infty}$ is a minimizer of the expected risk on $\mathcal{G}_\infty$. In this case we would have to balance the various terms in (11), which would lead to a worse bound. For example, we could consider $R := \log n$, obtaining a bound $n^{-1/2} \log n$ with an extra logarithmic term, but the result would hold only for $n$ larger than a number of examples $n_0$ at least *exponential* with respect to the norm of $f_\infty$. Moreover, to derive results uniform with respect to $f_\infty$, we would have to keep into account the decay rate of $A(R)$ and this would get bounds slower than $n^{-1/2}$.

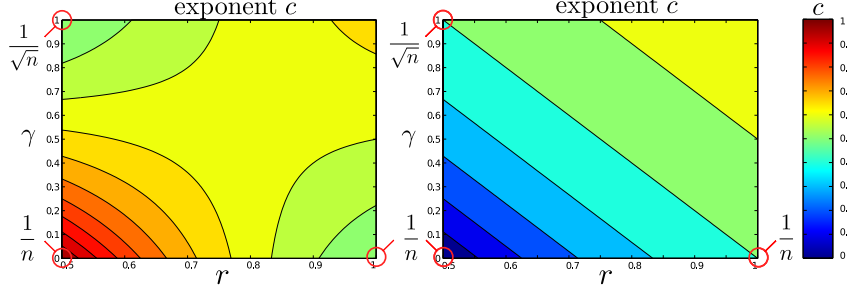

Figure 1: Random feat. $M = O(n^c)$ required for optimal generalization. Left: $\alpha = 1$. Right: $\alpha = \gamma$.

**Comparison with other results.**    Several other papers study the generalization properties of random features, see [22] and references therein. For example, generalization bounds are derived in [20] from very general arguments. However, the corresponding generalization bound requires a number of random features much larger than the number of training examples to give $O(1/\sqrt{n})$ bounds. The basic results in [22] are analogous to those in [19] with the set $\mathcal{G}_R$ replaced by $\mathcal{H}_R$. These results are closer, albeit more restrictive then ours (see Remark 8) and especially like the bounds in [19] suggest $O(n)$ random features are needed for $O(1/\sqrt{n})$ learning bounds. A novelty in [22] is the introduction of more complex problem dependent sampling that can reduce the number of random features. In Section 3.3, we show that using possibly-data dependent random features can lead to rates much faster than $n^{-1/2}$, and using much less than $\sqrt{n}$ features.

**Remark 3** (Sketching and randomized numerical linear algebra (RandLA))**.** *Standard sketching techniques from RandLA [24] can be recovered, when $X$ is a bounded subset of $\mathbb{R}^D$, by selecting $\psi(x, \omega) = x^\top \omega$ and $\omega$ sampled from suitable bounded distribution (e.g. $\omega = (\zeta_1, \ldots, \zeta_d)$ independent Rademacher random variables). Note however that the final goal of the analysis in the randomized numerical linear algebra community is to minimize the empirical error instead of $\mathcal{E}$.*

### 3.2   Refined Results: Fast Learning Rates

Faster rates can be achieved under favorable conditions. Such conditions for kernel ridge regression are standard, but somewhat technical. Roughly speaking they characterize the "size" of the considered RKHS and the regularity of $f_\mathcal{H}$. The key quantity needed to make this precise is the integral operator defined by the kernel $K$ and the marginal distribution $\rho_X$ of $\rho$ on $X$, that is

$$(Lg)(x) = \int_X K(x, z)g(z)d\rho_X(z), \quad \forall g \in L^2(X, \rho_X),$$

seen as a map from $L^2(X, \rho_X) = \{f : X \to \mathbb{R} \mid \|f\|_\rho^2 = \int |f(x)|^2 d\rho_X < \infty\}$ to itself. Under the assumptions of Thm. 1, the integral operator is positive, self-adjoint and trace-class (hence compact) [27]. We next define the conditions that will lead to fast rates, and then comment on their interpretation.

**Assumption 1** (Prior assumptions)**.** *For $\lambda > 0$, let the effective dimension be defined as $\mathcal{N}(\lambda) := \mathrm{Tr}\left((L + \lambda I)^{-1} L\right)$, and assume, there exists $Q > 0$ and $\gamma \in [0, 1]$ such that,*

$$\mathcal{N}(\lambda) \leq Q^2 \lambda^{-\gamma}. \tag{12}$$

*Moreover, assume there exists $r \geq 1/2$ and $g \in L^2(X, \rho_X)$ such that*

$$f_\mathcal{H}(x) = (L^r g)(x) \quad a.s. \tag{13}$$

We provide some intuition on the meaning of the above assumptions, and defer the interested reader to [10] for more details. The effective dimension can be seen as a "measure of the size" of the RKHS $\mathcal{H}$. Condition (12) allows to control the variance of the estimator and is equivalent to conditions on covering numbers and related capacity measures [26]. In particular, it holds if the eigenvalues $\sigma_i$'s of $L$ decay as $i^{-1/\gamma}$. Intuitively, a fast decay corresponds to a smaller RKHS, whereas a slow decay corresponds to a larger RKHS. The case $\gamma = 0$ is the more benign situation, whereas $\gamma = 1$ is the worst case, corresponding to the basic setting. A classic example, when $X = \mathbb{R}^D$, corresponds to

considering kernels of smoothness $s$, in which case $\gamma = D/(2s)$ and condition (12) is equivalent to assuming $\mathcal{H}$ to be a Sobolev space [26]. Condition (13) allows to control the bias of the estimator and is common in approximation theory [28]. It is a regularity condition that can be seen as form of weak sparsity of $f_{\mathcal{H}}$. Roughly speaking, it requires the expansion of $f_{\mathcal{H}}$, on the the basis given by the the eigenfunctions $L$, to have coefficients that decay faster than $\sigma_i^r$. A large value of $r$ means that the coefficients decay fast and hence many are close to zero. The case $r = 1/2$ is the worst case, and can be shown to be equivalent to assuming $f_{\mathcal{H}}$ exists. This latter situation corresponds to setting considered in the previous section. We next show how these assumptions allow to derive fast rates.

**Theorem 2.** *Let $\delta \in (0,1]$. Under Asm. 1 and the same assumptions of Thm. 1, if $n \geq n_0$, and $\lambda_n = n^{-\frac{1}{2r+\gamma}}$, then a number of random features $M$ equal to*

$$M_n = c_0\, n^{\frac{1+\gamma(2r-1)}{2r+\gamma}}\ \log \frac{108\kappa^2 n}{\delta},$$

*is enough to guarantee, with probability at least $1 - \delta$, that*

$$\mathcal{E}(\widehat{f}_{\lambda_n, M_n}) - \mathcal{E}(f_{\mathcal{H}}) \leq c_1 \log^2 \frac{18}{\delta}\ n^{-\frac{2r}{2r+\gamma}},$$

*for $r \leq 1$, and where $c_0, c_1$ do not depend on $n, \tau$, while $n_0$ does not depends on $n, f_{\mathcal{H}}, \rho$.*

The above bound is the same as the one obtained by the full kernel ridge regression estimator and is optimal in a minimax sense [10]. For large $r$ and small $\gamma$ it approaches a $O(1/n)$ bound. When $\gamma = 1$ and $r = 1/2$ the worst case bound of the previous section is recovered. Interestingly, the number of random features in different regimes is typically smaller than $n$ but can be larger than $O(\sqrt{n})$. Figure. 1 provides a pictorial representation of the number of random features needed for optimal rates in different regimes. In particular $M \ll n$ random features are enough when $\gamma > 0$ and $r > 1/2$. For example for $r = 1, \gamma = 0$ (higher regularity/sparsity and a small RKHS) $O(\sqrt{n})$ are sufficient to get a rate $O(1/n)$. But, for example, if $r = 1/2, \gamma = 0$ (not too much regularity/sparsity but a small RKHS) $O(n)$ are needed for $O(1/n)$ error. The proof suggests that this effect can be a byproduct of sampling features in a data-independent way. Indeed, in the next section we show how much fewer features can be used considering problem dependent sampling schemes.

### 3.3 Refined Results: Beyond uniform sampling

We show next that fast learning rates can be achieved with fewer random features if they are somewhat *compatible* with the data distribution. This is made precise by the following condition.

**Assumption 2** (Compatibility condition)**.** *Define the* maximum random features dimension *as*

$$\mathcal{F}_{\infty}(\lambda) = \sup_{\omega \in \Omega} \|(L + \lambda I)^{-1/2}\psi(\cdot, \omega)\|_{\rho_X}^2, \quad \lambda > 0. \tag{14}$$

*Assume there exists $\alpha \in [0,1]$, and $F > 0$ such that $\mathcal{F}_{\infty}(\lambda) \leq F\lambda^{-\alpha}, \quad \forall \lambda > 0$.*

The above assumption is abstract and we comment on it before showing how it affects the results. The maximum random features dimension (14) relates the random features to the data-generating distribution through the operator $L$. It is always satisfied for $\alpha = 1$ ands $F = \kappa^2$. e.g. considering any random feature satisfying (6). The favorable situation corresponds to random features such that case $\alpha = \gamma$. The following theoretical construction borrowed from [22] gives an example.

**Example 2** (Problem dependent RF)**.** *Assume $K$ is a kernel with an integral representation (6). For $s(\omega) = \|(L + \lambda I)^{-1/2}\psi(\cdot, \omega)\|_{\rho_X}^{-2}$ and $C_s := \int \frac{1}{s(\omega)}d\pi(\omega)$, consider the random features $\psi_s(x, \omega) = \psi(x, \omega)\sqrt{C_s s(\omega)}$, with distribution $\pi_s(\omega) := \frac{\pi(\omega)}{C_s s(\omega)}$. We show in the Appendix that these random features provide an integral representation of $K$ and satisfy Asm. 2 with $\alpha = \gamma$.*

We next show how random features satisfying Asm. 2 can lead to better resuts.

**Theorem 3.** *Let $\delta \in (0,1]$. Under Asm. 2 and the same assumptions of Thm. 1, 2, if $n \geq n_0$, and $\lambda_n = n^{-\frac{1}{2r+\gamma}}$, then a number of random features $M_n$ equal to*

$$M_n = c_0\, n^{\frac{\alpha+(1+\gamma-\alpha)(2r-1)}{2r+\gamma}}\ \log \frac{108\kappa^2 n}{\delta},$$

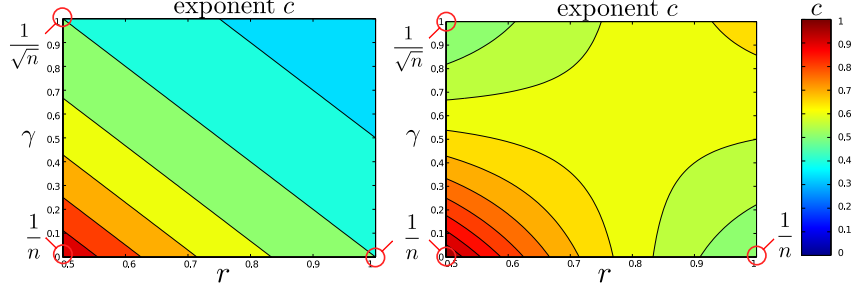

Figure 2: Comparison between the number of features $M = O(n^c)$ required by Nyström (uniform sampling, left) [16] and Random Features ($\alpha = 1$, right), for optimal generalization.

*is enough to guarantee, with probability at least $1 - \delta$, that*

$$\mathcal{E}(\widehat{f}_{\lambda_n, M_n}) - \mathcal{E}(f_\mathcal{H}) \leq c_1 \log^2 \frac{18}{\delta}\, n^{-\frac{2r}{2r+\gamma}},$$

*where $c_0, c_1$ do not depend on $n, \tau$, while $n_0$ does not depends on $n, f_\mathcal{H}, \rho$.*

The above learning bound is the same as Thm. 2, but the number of random features is given by a more complex expression depending on $\alpha$. In particular, in the slow $O(1/\sqrt{n})$ rates scenario, that is $r = 1/2, \gamma = 1$, we see that $O(n^{\alpha/2})$ are needed, recovering $O(\sqrt{n})$, since $\gamma \leq \alpha \leq 1$. On the contrary, for a small RKHS, that is $\gamma = 0$ and random features with $\alpha = \gamma$, a constant (!) number of feature is sufficient. A similar trend is seen considering fast rates. For $\gamma > 0$ and $r > 1/2$, if $\alpha < 1$ then the number of random features is always smaller, and potentially much smaller, then the number of random features sampled in a problem independent way, that is $\alpha = 1$. For $\gamma = 0$ and $r = 1/2$, the number of number of features is $O(n^\alpha)$ and can be again just constant if $\alpha = \gamma$. Figure 1 depicts the number of random features required if $\alpha = \gamma$. The above result shows the potentially dramatic effect of problem dependent random features. However the construction in Ex. 2 is theoretical. We comment on this in the next remark.

**Remark 4** (Random features leverage scores)**.** *The construction in Ex. 2 is theoretical, however empirical random features leverage scores $\widehat{s}(\omega) = \widehat{v}(\omega)^\top (\mathbf{K} + \lambda n I)^{-1} \widehat{v}(\omega)$, with $\widehat{v}(\omega) \in \mathbb{R}^n$, $(\widehat{v}(\omega))_i = \psi(x_i, \omega)$, can be considered. Statistically, this requires considering an extra estimation step. It seems our proof can be extended to account for this, and we will pursue this in a future work. Computationally, it requires devising approximate numerical strategies, like standard leverage scores* [23].

**Comparison with Nyström.**   This question was recently considered in [21] and our results offer new insights. In particular, recalling the results in [16], we see that in the slow rate setting there is essentially no difference between random features and Nyström approaches, neither from a statistical nor from a computational point of view. In the case of fast rates, Nyström methods with uniform sampling requires $O(n^{-\frac{1}{2r+\gamma}})$ random centers, which compared to Thm. 2, suggests Nyström methods can be advantageous in this regime. While problem dependent random features provide a further improvement, it should be compared with the number of centers needed for Nyström with leverage scores, which is $O(n^{-\frac{\gamma}{2r+\gamma}})$ and hence again better, see Thm. 3. In summary, both random features and Nyström methods achieve optimal statistical guarantees while reducing computations. They are essentially the same in the worst case, while Nyström can be better for benign problems.
Finally we add a few words about the main steps in the proof.

**Steps of the proof.**   The proofs are quite technical and long and are collected in the appendices. They use a battery of tools developed to analyze KRR and related methods. The key challenges in the analysis include analyzing the bias of the estimator, the effect of noise in the outputs, the effect of random sampling in the data, the approximation due to random features and a notion of orthogonality between the function space corresponding to random features and the full RKHS. The last two points are the main elements on novelty in the proof. In particular, compared to other studies, we identify and study the quantity needed to assess the effect of the random feature approximation if the goal is prediction rather than the kernel approximation itself.

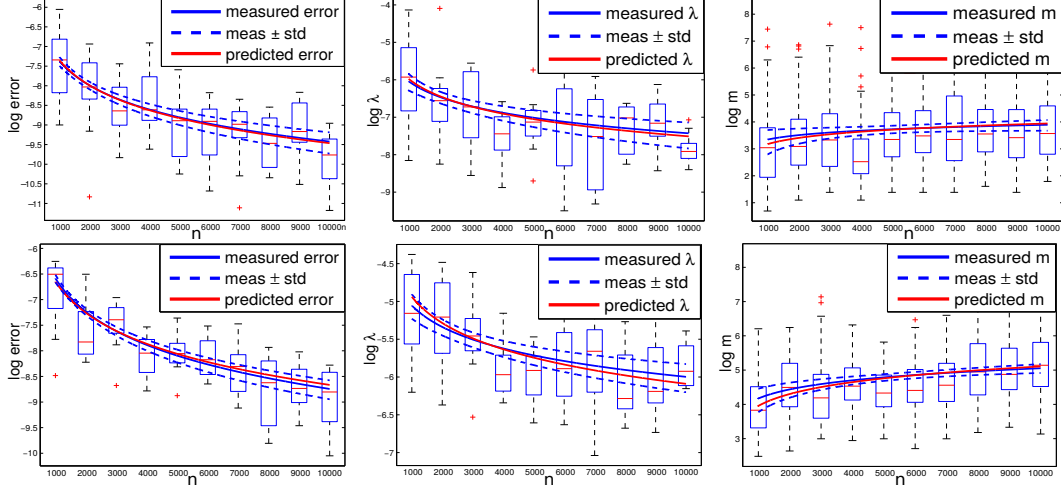

Figure 3: Comparison of theoretical and simulated rates for: excess risk $\mathcal{E}(\widehat{f}_{\lambda,M}) - \inf_{f \in \mathcal{H}} \mathcal{E}(f)$, $\lambda$, $M$, w.r.t. $n$ (100 repetitions). Parameters $r = 11/16, \gamma = 1/8$ (top), and $r = 7/8, \gamma = 1/4$ (bottom).

## 4 Numerical results

While the learning bounds we present are optimal, there are no lower bounds on the number of random features, hence we present numerical experiments validating our bounds. Consider a spline kernel of order $q$ (see [29] Eq. 2.1.7 when $q$ integer), defined as $\Lambda_q(x, x') = \sum_{k=-\infty}^{\infty} e^{2\pi i k x} e^{-2\pi i k z} |k|^{-q}$, almost everywhere on $[0, 1]$, with $q \in \mathbb{R}$, for which we have $\int_0^1 \Lambda_q(x, z) \Lambda_{q'}(x', z) dz = \Lambda_{q+q'}(x, x')$, for any $q, q' \in \mathbb{R}$. Let $X = [0, 1]$, and $\rho_X$ be the uniform distribution. For $\gamma \in (0, 1)$ and $r \in [1/2, 1]$ let, $K(x, x') = \Lambda_{\frac{1}{\gamma}}(x, x')$, $\psi(\omega, x) = \Lambda_{\frac{1}{2\gamma}}(\omega, x)$, $f_*(x) = \Lambda_{\frac{r}{\gamma} + \frac{1}{2} + \epsilon}(x, x_0)$ with $\epsilon > 0, x_0 \in X$. Let $\rho(y|x)$ be a Gaussian density with variance $\sigma^2$ and mean $f^*(x)$. Then Asm 1, 2 are satisfied and $\alpha = \gamma$. We compute the KRR estimator for $n \in \{10^3, \ldots, 10^4\}$ and select $\lambda$ minimizing the excess risk computed analytically. Then we compute the RF-KRR estimator and select the number of features $M$ needed to obtain an excess risk within $5\%$ of the one by KRR. In Figure 3, the theoretical and estimated behavior of the excess risk, $\lambda$ and $M$ with respect to $n$ are reported together with their standard deviation over 100 repetitions. The experiment shows that the predictions by Thm. 3 are accurate, since the theoretical predictions estimations are within one standard deviation from the values measured in the simulation.

## 5 Conclusion

In this paper, we provide a thorough analyses of the generalization properties of random features with ridge regression. We consider a statistical learning theory setting where data are noisy and sampled at random. Our main results show that there are large classes of learning problems where random features allow to reduce computations while preserving optimal statistical accuracy of exact kernel ridge regression. This in contrast with previous state of the art results suggesting computational gains needs to be traded-off with statistical accuracy. Our results open several venues for both theoretical and empirical work. As mentioned in the paper, it would be interesting to analyze random features with empirical leverage scores. This is immediate if input points are fixed, but our approach should allow to also consider the statistical learning setting. Beyond KRR, it would be interesting to analyze random features together with other approaches, in particular accelerated and stochastic gradient methods, or distributed techniques. It should be possible to extend the results in the paper to consider these cases. A more substantial generalization would be to consider loss functions other than quadratic loss, since this require different techniques from empirical process theory.

**Acknowledgments**   The authors gratefully acknowledge the contribution of Raffaello Camoriano who was involved in the initial phase of this project. These preliminary result appeared in the 2016 NIPS workshop "Adaptive and Scalable Nonparametric Methods in ML". This work is funded by the Air Force project FA9550-17-1-0390 (European Office of Aerospace Research and Development) and by the FIRB project RBFR12M3AC (Italian Ministry of Education, University and Research).

## Footnotes

* This work was done when A.R. was working at Laboratory of Computational and Statistical Learning (Istituto Italiano di Tecnologia).

[2]A kernel $K$ is PD if for all $x_1, \ldots, x_N$ the $N$ by $N$ matrix with entries $K(x_i, x_j)$ is positive semidefinite.

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
