[Supplementary Material]

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

# Generalization Properties of Learning with Random Features
## Supplementary Materials

The supplementary materials are divided in the following four section

*A. Proofs* - where the proofs for Section 3 are provided

*B. Concentration Inequalities* - where probabilistic tools necessary for the proofs are recalled

*C. Operator Inequalities* - where some analytic inequalities used in the proofs are recalled

*D. Auxiliary Results* - where some technical lemmas necessary to the proof are derived

*E. Examples of Random Features* - where examples of random features expansion are recalled

## A  Proofs

In Sect. A.1, the notation is introduced and some standard identities are recalled. In Sect. A.2, the excess risk is decomposed in five terms (Eq. (17)-(21)) that are further simplified in Lemma 2, 3, 4, 5. The complete decomposition is presented in Thm. 4. In Sect. A.3, the terms in decomposition are bounded in probability, in particular Lemma 7 bounds the variance term, Lemma 8 the computational error term, while Lemma 10 controls the constants. Finally the proofs of the main results are presented in Section A.4 together with the more general results of Thm. 5.

First we recall the assumptions needed to derive the results. They are already presented or implied in the main text, here we collect and number them.

**Assumption 2** (Compatibility condition) *There exists $\alpha \in [0,1]$ and $F > 0$ such that*

$$\mathcal{F}_\infty(\lambda) \leq F\lambda^{-\alpha}, \quad \forall \lambda > 0.$$

**Assumption 3** (Random Features are bounded and continuous). *The kernel $K$ has an integral representation as in Eq. 6, with $\psi$ continuous in both variables and bounded, that is, there exists $\kappa \geq 1$ such that $|\psi(x,\omega)| \leq \kappa$ for any $x, \in X$ and $\omega \in \Omega$. The associated RKHS $\mathcal{H}$ is separable.*

Note that the assumption above is satisfied when the random feature is continuous and bounded and the space $X$ is separable (e.g. $\mathbb{R}^d$, $d \in \mathbb{N}$ or any Polish space). Indeed the continuity of $\psi$ implies the continuity of $K$, which, together with the separability of $X$ implies the separability of $\mathcal{H}$.

**Assumption 4** (Noise on the $y$ is sub-exponential, and there exists $f_\mathcal{H}$). *For any $x \in X$*

$$\mathbb{E}[|y|^p \mid x] \leq \frac{1}{2}p!\sigma^2 B^{p-2}, \quad \forall p \geq 2.$$

*Moreover there exists $f_\mathcal{H} \in \mathcal{H}$ such that $\mathcal{E}(f_\mathcal{H}) = \inf_{f \in \mathcal{H}} \mathcal{E}(f)$.*

Note that the above assumption on $y$ is satisfied when $y$ is bounded, sub-gaussian or sub-exponential. In particular, if $|y| \in [-\frac{b}{2}, \frac{b}{2}]$ almost surely, with $b \in (0, \infty)$ then the assumption above is satisfied with $\sigma = B = b$.

**Assumption 5** (Effective dimension). *Let $\lambda > 0$. There exists $Q > 0$ and $\gamma \in [0,1]$ such that, for any $\lambda > 0$*

$$\mathcal{N}(\lambda) \leq Q^2 \lambda^{-\gamma}.$$

It is the first part of Asm. 1, for the sake of clarity we need to split it in two, since many results depend either on the first or on the second part.

**Assumption 6** (Source condition). *There exists $1/2 \leq r \leq 1$ and $g \in L^2(X, \rho_X)$ such that*

$$f_\mathcal{H}(x) = (L^r g)(x) \quad a.s.$$

*We denote with $R$ the quantity $1 \vee \|g\|_{\rho_X}$.*

## A.1 Kernel and Random Features Operators

In this section, we provide the notation, recall some useful facts and define some operators used in the rest of the appendix. In the rest of the paper we denote with $\|\cdot\|$ the operatorial norm and with $\|\cdot\|_{HS}$ the Hilbert-Schmidt norm. Let $\mathcal{L}$ be a Hilbert space, we denote with $\langle\cdot,\cdot\rangle_{\mathcal{L}}$ the associated inner product, with $\|\cdot\|_{\mathcal{L}}$ the norm and with $\mathrm{Tr}(\cdot)$ the trace. Let $Q$ be a bounded self-adjoint linear operator on a separable Hilbert space $\mathcal{L}$, we denote with $\lambda_{\max}(Q)$ the biggest eigenvalue of $Q$, that is $\lambda_{\max}(Q) = \sup_{\|f\|_{\mathcal{L}} \leq 1} \langle f, Qf\rangle_{\mathcal{L}}$. Moreover, we denote with $Q_\lambda$ the operator $Q + \lambda I$, where $Q$ is a linear operator, $\lambda \in \mathbb{R}$ and $I$ the identity operator, so for example $\widehat{C}_{M,\lambda} := \widehat{C}_M + \lambda I$. Moreover we recall some basic properties of norms in Hilbert spaces.

**Remark 5.** *Let $V_0, \ldots, V_t$ with $t \in \mathbb{N}$ be Hilbert spaces. Let $q \in V_0$ and $A_i : V_i \to V_{i-1}$ bounded linear operators and $f \in V_t$. We recall that the identity $q = (A_1)\cdots(A_t)(f)$, implies $\|q\|_{V_0} \leq \|A_1\|\ldots\|A_t\|\|f\|_{V_t}$.*

Let $X$ be a probability space and $\rho$ be a probability distribution on $X \times \mathbb{R}$ satisfying Assumption 3. We denote $\rho_X$ its marginal on $X$ and $\rho(y|x)$ the conditional distribution on $\mathbb{R}$. Let $L^2(X, \rho_X)$ be the Lebesgue space of square $\rho_X$-integrable functions, with the canonical inner product

$$\langle g, h\rangle_{\rho_X} = \int_X g(x)h(x)d\rho_X(x), \quad \forall g, h \in L^2(X, \rho_X),$$

and the norm $\|g\|_{\rho_X}^2 = \langle g, g\rangle_{\rho_X}$, for all $g \in L^2(X, \rho_X)$. Let $(\Omega, \pi)$ be a probability space and $\psi : \Omega \times X \to \mathbb{R}$ be a continuous and bounded map as in Asm. 3. Moreover let the kernel $K$ be defined by Eq. (6). We denote with $K_x$ the function $K(x, \cdot)$, for any $x \in X$. Then the Reproducing Kernel Hilbert Space $\mathcal{H}$ induced by $K$ is defined by

$$\mathcal{H} = \overline{\mathrm{span}\{K_x \mid x \in X\}}, \quad \text{completed with} \quad \langle K_x, K_{x'}\rangle_{\mathcal{H}} = K(x, x') \ \forall x, x' \in X.$$

We now define the operators needed in the rest of the proofs. Let $n \in \mathbb{N}$, and $(x_1, y_1), \ldots, (x_n, y_n) \in X \times \mathbb{R}$ be sampled independently according to $\rho$.

**Definition 1.** *Let $P : L^2(X, \rho_X) \to L^2(X, \rho_X)$ be the projection operator with the same range of $L$. Let $f_\rho : X \to \mathbb{R}$ be defined as*

$$f_\rho(x) = \int y d\rho(y|x) \ \ a.\,e.$$

We now recall a useful characterization of the excess risk, in term of the quantities defined above.

**Remark 6** (from [2, 30]). *When $\int y^2 d\rho$ is finite, then $f_\rho \in L^2(X, \rho_X)$ and $f_\rho$ is the minimizer of $\mathcal{E}$ over all the measurable functions. When $\int K(x, x)d\rho_X$ is finite, the range of $P$ is the closure of $\mathcal{H}$ in $L^2(X, \rho_X)$, that this the closure of the range of $L$. When both conditions hold, for any $f \in L^2(X, \rho_X)$ the following hold*

$$\mathcal{E}(f) - \inf_{g \in \mathcal{H}} \mathcal{E}(g) = \|f - Pf_\rho\|_{\rho_X}^2.$$

*Moreover if there exists $f_{\mathcal{H}} \in \mathcal{H}$ minimizing $\mathcal{E}$, then Asm. 6 is equivalent to requiring the existence of $r \geq 1/2$, $g \in L^2(X, \rho_X)$ such that*

$$Pf_\rho = L^r g, \tag{15}$$

*with $R := \|g\|_{L^2(X,\rho_X)}$.*

In the following we define analogous operators for the approximated kernel $K_M := \phi_M(x)^\top \phi_M(x')$, with

$$\phi_M(x) := M^{-1/2}(\psi(x, \omega_1), \ldots, \psi(x, \omega_M)),$$

for any $x, x' \in X$, where $M \in \mathbb{N}$ and $\omega_1, \ldots, \omega_M \in \Omega$ are sampled independently according to $\pi$. We denote with $\psi_\omega$ the function $\psi(\cdot, \omega)$ for any $\omega \in \Omega$. According to the following remark, we have that $\psi_{\omega_i} \in L^2(X, \rho_X)$ almost surely.

**Remark 7.** *Under Asm. 3 and the fact that $\rho$ is a finite measure, $\psi_\omega \in L^2(X, \rho_X)$ almost surely.*

Now we are ready for defining the following operators, depending on $\phi_M$ or $K_M$.

**Definition 2.** *For all $g \in L^2(X, \rho_X)$, $\beta \in \mathbb{R}^M$, $\alpha \in \mathbb{R}^n$ and $i \in \{1, \dots, M\}$, we have*

- $S_M : \mathbb{R}^M \to L^2(X, \rho_X), \quad (S_M\beta)(\cdot) = \phi_M(\cdot)^\top \beta$,

- $S_M^* : L^2(X, \rho_X) \to \mathbb{R}^M, \quad (S_M^* g)_i = \frac{1}{\sqrt{M}} \int_X \psi_{\omega_i}(x) g(x) d\rho_X(x)$,

- $L_M : L^2(X, \rho_X) \to L^2(X, \rho_X), \quad (L_M g)(\cdot) = \int_X K_M(\cdot, z) g(z) d\rho_X(z)$.

- $C_M : \mathbb{R}^M \to \mathbb{R}^M, \quad C_M = \int_X \phi_M(x)\phi_M(x)^\top d\rho_X(x)$,

- $\widehat{C}_M : \mathbb{R}^M \to \mathbb{R}^M, \quad \widehat{C}_M = \frac{1}{n} \sum_{i=1}^n \phi_M(x_i)\phi_M(x_i)^\top$.

Note that the operators above satisfy the properties in the following remark.

**Remark 8** (from [10])**.** *Under Asm. 3 the linear operators $L$ is trace class and $L_M, C_M, S_M, \widehat{C}_M, \widehat{S}_M$ are finite dimensional. Moreover we have that $L = SS^*$, $L_M = S_M S_M^*$, $C_M = S_M^* S_M$ and $\widehat{C}_M = \widehat{S}_M^* \widehat{S}_M$. Finally $L, L_M, C_M, \widehat{C}_M$ are self-adjoint and positive operators, with spectrum is $[0, \kappa^2]$.*

In the next remark we rewrite $\widehat{f}_{\lambda,M}$ in terms of the operators introduced above.

**Remark 9.** *Let $\widehat{f}_{\lambda,M}$ defined as in Eq. 7. Under Assumption 3, $\widehat{f}_{\lambda,M} \in L^2(X, \rho_X)$ almost surely, since $\psi_\omega$ is in $L^2(X, \rho_X)$ almost surely (Rem. 7) and $\widehat{f}_{\lambda,M}$ is a linear combination of $\psi_{\omega_1}, \dots, \psi_{\omega_M}$. In particular,*

$$\widehat{f}_{\lambda,M} = S_M \widehat{C}_{M,\lambda}^{-1} \widehat{S}_M^* \widehat{y}.$$

## A.2 Analytic Result

In this subsection we decompose analytically the excess risks in different terms, that will be bounded via concentration inequalities in the next section. Under Asm. 3, since $\widehat{f}_{\lambda,M} \in L^2(X, \rho_X)$ almost surely, we have

$$\mathcal{E}(\widehat{f}_{\lambda,M}) - \inf_{f \in \mathcal{H}} \mathcal{E}(f) = \|\widehat{f}_{\lambda,M} - Pf_\rho\|_{\rho_X}^2, \tag{16}$$

(for more details see Rem. 6, 9). In our analysis we decompose the excess risk in the following five terms

$$\widehat{f}_{\lambda,M} - Pf_\rho = \widehat{f}_{\lambda,M} - S_M \widehat{C}_{M,\lambda}^{-1} S_M^* f_\rho \tag{17}$$

$$+ \ S_M \widehat{C}_{M,\lambda}^{-1} S_M^* (I - P) f_\rho \tag{18}$$

$$+ \ S_M \widehat{C}_{M,\lambda}^{-1} S_M^* P f_\rho - L_M L_{M,\lambda}^{-1} P f_\rho \tag{19}$$

$$+ \ L_M L_{M,\lambda}^{-1} P f_\rho - L L_\lambda^{-1} P f_\rho \tag{20}$$

$$+ \ L L_\lambda^{-1} P f_\rho - P f_\rho. \tag{21}$$

The first controls the variance of the outputs $y$, the second the interaction between the space of models spanned by $\psi_{\omega_1}, \dots, \psi_{\omega_M}$ and $\mathcal{H}$, the third the approximation of the inverse covariance operator $\widehat{C}_{M,\lambda}^{-1}$, the fourth controls how close is the integral operator $L_M$ to $L$, while the last controls the approximation error of the models in $\mathcal{H}$. The $L^2(X, \rho_X)$ norm of $\widehat{f}_{\lambda,M} - Pf_\rho$ is bounded by the sum of the $L^2(X, \rho_X)$ norms of the terms, that are further bounded in Lemma. 2, 3, 4, 5. The final analytical decomposition is given in Thm. 4. First we need a preliminary result.

**Lemma 1.** *Under Asm. 3, the operator $L$ is characterized by*

$$L = \int \psi_\omega \otimes \psi_\omega d\pi(\omega).$$

*Proof.* By Asm. 3, we have that $\psi_\omega \in L^2(X, \rho_X)$ almost surely and uniformly bounded. By using the kernel expansion of Eq. (6), the linearity of the Bochner integral and of the dot product, we have

that for any $f, g \in L^2(X, \rho_X)$ the following holds

$$
\begin{aligned}
\langle f, Lg \rangle_{\rho_X} &= \int f(x)K(x,z)g(z)d\rho_X(x)d\rho_X(z) \\
&= \int f(x)\psi(x,\omega)\psi(z,\omega)g(z)d\rho_X(x)d\rho_X(z)d\pi(\omega) \\
&= \int \langle f, \psi_\omega \rangle_{\rho_X} \langle g, \psi_\omega \rangle_{\rho_X} \, d\pi(\omega) = \left\langle f, \int \psi_\omega \langle g, \psi_\omega \rangle_{\rho_X} \, d\pi(\omega) \right\rangle_{\rho_X} \\
&= \left\langle f, \left( \int \psi_\omega \otimes \psi_\omega d\pi(\omega) \right) g \right\rangle_{\rho_X}.
\end{aligned}
$$

$\square$

Now we are ready to prove that the second term of the expansion in Eq. 17 is zero. We obtain this result by proving that $\|(I - P)\psi_\omega\| = 0$ almost everywhere.

**Lemma 2.** *Under Asm. 3, the following holds for any $\lambda > 0, M, n \in \mathbb{N}$,*

$$
\|S_M \widehat{C}_{M,\lambda}^{-1} S_M^* (I - P)f_\rho\|_{\rho_X} = 0 \ a.s.
$$

*Proof.* Note that, since $P$ is the projection operator on the closure of the range of $L$ and $L$ is trace class, then $(I - P)L = 0$, this implies that $\text{Tr}((I - P)L(I - P)) = 0$. By the characterization of $L$ given in Lemma 1, the linearity of the bounded operator $I - P$ and of the trace, we have that

$$
\begin{aligned}
0 = \text{Tr}((I - P)L(I - P)) &= \text{Tr}\left( (I - P)\left( \int \psi_\omega \otimes \psi_\omega d\pi(\omega) \right)(I - P) \right) \\
&= \int \text{Tr}((I - P)(\psi_\omega \otimes \psi_\omega)(I - P)) \, d\pi(\omega) \\
&= \int \|(I - P)\psi_\omega\|_{\rho_X}^2 \, d\pi(\omega),
\end{aligned}
$$

where the last step is due to the fact that $\text{Tr}(A(v \otimes v)A) = \text{Tr}(Av \otimes Av) = \|Av\|_{\rho_X}^2$ for any bounded self adjoint operator $A$ and any function $v \in L^2(X, \rho_X)$. The equation above implies that $(I - P)\psi_\omega = 0$ almost surely on the support of $\pi$. Now we study $S_M^*(I - P)$, for any $\beta \in \mathbb{R}^M$ and any $f \in L^2(X, \rho_X)$ we have

$$
\langle \beta, S_M^*(I - P)f \rangle_{\mathbb{R}^M} = \frac{1}{\sqrt{M}} \sum_{i=1}^M \beta_i \langle (I - P)\psi_{\omega_i}, f \rangle_{\rho_X} = 0 \ a.s.,
$$

where the last step is due to the fact that $\langle 0, v \rangle = 0$ for any $v$ and $(I - P)\psi_{\omega_i} = 0$ almost surely, since $\omega_i$ are distributed according to $\pi$ and $(I - P)\psi_\omega = 0$ almost surely on the support of $\pi$. Now

$$
\|S_M \widehat{C}_{M,\lambda}^{-1} S_M^* (I - P)f_\rho\|_{\rho_X} \leq \|S_M \widehat{C}_{M,\lambda}^{-1}\|\|S_M^*(I - P)\|\|f_\rho\|_{\rho_X} = 0 \ a.s.
$$

$\square$

**Lemma 3.** *Under Asm. 3, and Eq. (15) the following holds for any $\lambda > 0, M, n \in \mathbb{N}$*

$$
\|S_M \widehat{C}_{M,\lambda}^{-1} S_M^* P f_\rho - L_M L_{M,\lambda}^{-1} P f_\rho\| \leq R\kappa^{2r-1}\|L_{M,\lambda}^{-1/2}L^{1/2}\|\|S_M \widehat{C}_{M,\lambda}^{-1} C_{M,\lambda}^{-1/2}\|\|C_{M,\lambda}^{-1/2}(C_M - \widehat{C}_M)\|.
$$

*Proof.* First of all we recall that $Z^* f(ZZ^*) = f(Z^*Z)Z^*$ for any continuous spectral function and any compact operator $Z$. By the characterization of $L_M$ in Rem. 8 under Asm. 3, we have

$$
L_M L_{M,\lambda}^{-1} = S_M S_M^* (S_M S_M^* + \lambda I)^{-1} = S_M (S_M^* S_M + \lambda I)^{-1} S_M^* = S_M C_{M,\lambda}^{-1} S_M^*,
$$

since $(\cdot + \lambda I)^{-1}$ is a continuos spectral function on $[0, \infty)$, which contains the spectrum of $L$ that is in $[0, \kappa^2]$. Equivalently, the equation above could be proven algebraically via the Woodbury identity. Now we have

$$
(S_M \widehat{C}_{M,\lambda}^{-1} S_M^* - L_M L_{M,\lambda}^{-1}) P f_\rho = S_M (\widehat{C}_{M,\lambda}^{-1} - C_{M,\lambda}^{-1}) S_M^* P f_\rho = S_M \widehat{C}_{M,\lambda}^{-1} (C_M - \widehat{C}_M) C_{M,\lambda}^{-1} S_M^* P f_\rho,
$$

where the last step is due to the identity $A^{-1} - B^{-1} = A^{-1}(B - A)B^{-1}$ valid for any bounded invertible linear operator $A, B$. In particular by multiplying and dividing by $C_{M,\lambda}^{1/2}$ we have the following decomposition

$$S_M \widehat{C}_{M,\lambda}^{-1}(C_M - \widehat{C}_M)C_{M,\lambda}^{-1}S_M^* Pf_\rho = (S_M \widehat{C}_{M,\lambda}^{-1} C_{M,\lambda}^{1/2}) \ (C_{M,\lambda}^{-1/2}(C_M - \widehat{C}_M)) \ (C_{M,\lambda}^{-1}S_M^* Pf_\rho).$$

The result is given by bounding the norm of the lhs of the identity above, by the product of the norms of the parentheses on the rhs (see Rem. 5). Note that by applying Eq. (15), we have that there exists $g \in L^2(X, \rho_X)$, such that $Pf_\rho = L^r g$ and by dividing and multiplying for $L_{M,\lambda}^{1/2}$, we have

$$C_{M,\lambda}^{-1}S_M^* Pf_\rho = (C_{M,\lambda}^{-1}S_M^* L_{M,\lambda}^{1/2}) \ (L_{M,\lambda}^{-1/2}L^{1/2}) \ L^{r-1/2} \ g.$$

Now note that, by Asm. 6, we have $r \geq 1/2$, $\|g\|_{\rho_X} \leq R$ and $\|L^{r-1/2}\| \leq \kappa^{2r-1}$ since $L$ is compact with the spectrum in $[0, \kappa^2]$ and $2r - 1 \geq 0$. By the fact that $(\cdot + \lambda I)^{-2}$ is a continuous spectral function on $[0, \infty)$ containing the spectrum of $C_M$, we have that $S_M C_{M,\lambda}^{-2} S_M^* = L_{M,\lambda}^{-2} L_M$ and so for any $\lambda > 0$

$$\|C_{M,\lambda}^{-1}S_M^* L_M^{1/2}\|^2 = \|L_M^{1/2} S_M C_{M,\lambda}^{-2} S_M^* L_M^{1/2}\| = \|L_{M,\lambda}^{-2} L_M^2\| \leq 1.$$

$\square$

**Lemma 4.** *Under Asm. 3, and Eq.* (15) *the following holds for any* $\lambda > 0, M \in \mathbb{N}$

$$\|(LL_\lambda^{-1} - L_M L_{M,\lambda}^{-1})Pf_\rho\| \leq R\sqrt{\lambda}\|L_{M,\lambda}^{-1/2}L_\lambda^{1/2}\|\|L_\lambda^{-1/2}(L - L_M)\|^{2r-1}\|L_\lambda^{-1/2}(L - L_M)L_\lambda^{-1/2}\|^{2-2r}$$

*Proof.* By the algebraic identities $A(A + \lambda I) = I - \lambda(A + \lambda I)^{-1}$ valid for any bounded positive operator and $A^{-1} - B^{-1} = A^{-1}(B - A)B^{-1}$ valid for any invertible bounded operators, we have

$$(LL_\lambda^{-1} - L_M L_{M,\lambda}^{-1})Pf_\rho = \lambda(L_{M,\lambda}^{-1} - L_\lambda^{-1})Pf_\rho = \lambda L_{M,\lambda}^{-1}(L - L_M)L_\lambda^{-1}Pf_\rho.$$

By applying Eq. (15), we have that there exists $g \in L^2(X, \rho_X)$, such that $Pf_\rho = L^r g$, so by multiplying and dividing by $L_M^{1/2}$, we preform the following decomposition

$$(LL_\lambda^{-1} - L_M L_{M,\lambda}^{-1})Pf_\rho = \sqrt{\lambda} \ (\sqrt{\lambda}L_{M,\lambda}^{-1/2}) \ (L_{M,\lambda}^{-1/2}L_\lambda^{1/2}) \ (L_\lambda^{-1/2}(L - L_M)L_\lambda^{-(1-r)}) \ (L_\lambda^{-r}L^r) \ g.$$

The result is given by bounding the norm of the lhs of the identity above, by the product of the norms of the parentheses on the rhs (see Rem. 5). Note that $\|\sqrt{\lambda}L_{M,\lambda}^{-1/2}\| \leq 1$ and $\|L_\lambda^{-r}L^r\| \leq 1$ for any $\lambda > 0$ and $\|g\|_{\rho_X} \leq R$. Now we apply Proposition 9 on $\|L_\lambda^{-1/2}(L - L_M)L_\lambda^{-(1-r)}\|$, indeed note that $0 \leq 1 - r \leq 1/2$, so by setting $\sigma = 2 - 2r$, $X = L_\lambda^{-1/2}(L - L_M)$, $A = L_\lambda^{-1/2}$ and applying the proposition, we have

$$\|L_\lambda^{-1/2}(L - L_M)L_\lambda^{-\sigma/2}\| \leq \|L_\lambda^{-1/2}(L - L_M)\|^{2r-1}\|L_\lambda^{-1/2}(L - L_M)L_\lambda^{-1/2}\|^{2-2r}.$$

$\square$

**Lemma 5.** *Under Asm. 3, and Eq.* (15) *the following holds for any* $\lambda > 0$,

$$\|LL_\lambda^{-1}Pf_\rho - Pf_\rho\| \leq R\lambda^r.$$

*Proof.* By the identity $A(A + \lambda I)^{-1} = I - \lambda(A + \lambda)^{-1}$ valid for $\lambda > 0$ and any bounded self-adjoint positive operator and by Eq. (15) for which there exists $g \in L^2(X, \rho_X)$ such that $Pf_\rho = L^r g$, we have

$$(I - LL_\lambda^{-1})Pf_\rho = \lambda L_\lambda^{-1}Pf_\rho = \lambda L_\lambda^{-1}L^r g = \lambda^r \ (\lambda^{1-r}L_\lambda^{-(1-r)}) \ (L_\lambda^{-r}L^r) \ g. \qquad (22)$$

The result is given by bounding the norm of the lhs of Eq. 22, by the product of the norms of the parentheses on the rhs (see Rem. 5). Note that $\|\lambda^{1-r}L_\lambda^{-(1-r)}\| \leq 1$ and $\|L_\lambda^{-r}L^r\| \leq 1$, while $R := \|g\|_{\rho_X}$ according to Eq. (15). $\square$

**Theorem 4** (Analytic Decomposition). *Under Assumptions 3 and Eq.* (15) *let* $\widehat{f}_{\lambda,M}$ *as in Eq. 7. For any $\lambda > 0$ and $M \in \mathbb{N}$, the following holds*

$$|\mathcal{E}(\widehat{f}_{\lambda,M}) - \inf_{f \in \mathcal{H}} \mathcal{E}(f)|^{1/2} \leq \beta ( \underbrace{\mathcal{S}(\lambda, M, n)}_{\text{Sample Error}} + \underbrace{\mathcal{C}(\lambda, M)}_{\text{Computational Error}} + \underbrace{R\lambda^v}_{\text{Approximation Error}} ) \quad (23)$$

*where $v = \min(r, 1)$,*

1. $\mathcal{S}(\lambda, M, n) := \|C_{M,\lambda}^{-1/2}(\widehat{S}_M^* \widehat{y} - S_M^* f_\rho)\|_{\rho X} + R\kappa^{2r-1}\|C_{M,\lambda}^{-1/2}(C_M - \widehat{C}_M)\|$,

2. $\mathcal{C}(\lambda, M) := R\sqrt{\lambda}\|L_\lambda^{-1/2}(L - L_M)\|^{2v-1}\|L_\lambda^{-1/2}(L - L_M)L_\lambda^{-1/2}\|^{2-2v}$,

3. $\beta := \max(1, (1 - \beta_1)^{-1})\max(1, (1 - \beta_2)^{-1/2})$, *with* $\beta_1 := \lambda_{\max}(C_{M,\lambda}^{-1/2}(C_M - \widehat{C}_M)C_{M,\lambda}^{-1/2})$ *and* $\beta_2 := \lambda_{\max}(L_\lambda^{-1/2}(L - L_M)L_\lambda^{-1/2})$.

*Proof.* Under Asm. 3, the excess risk is characterized by Eq. 16 as we recalled at the beginning of this subsection. We decomposed the quantity $\widehat{f}_{\lambda,M} - Pf_\rho$ according to the terms in Eq. 17-21. The $L^2(X, \rho_X)$ norm of $\widehat{f}_{\lambda,M} - Pf_\rho$ is bounded by the sum of the $L^2(X, \rho_X)$ norms of the terms, that are further bounded in Lemma. 2, 3, 4, 5. In particular, for the first term, by writing $\widehat{f}_{\lambda,M}$ in terms of the linear operators in Def. 2 (see Rem. 9) and by multipling and dividing by $C_{M,\lambda}^{1/2}$ we have

$$\widehat{f}_{\lambda,M} - S_M\widehat{C}_{M,\lambda}^{-1}S_M^* f_\rho = S_M\widehat{C}_{M,\lambda}^{-1}(\widehat{S}_M^* \widehat{y} - S_M^* f_\rho) = (S_M\widehat{C}_{M,\lambda}^{-1}C_{M,\lambda}^{1/2})(C_{M,\lambda}^{-1/2}(\widehat{S}_M^* \widehat{y} - S_M^* f_\rho)),$$

then we bound the norm of the term with the norm of the parenthesis in the decomposition above (see Rem. 5). By collecting the result above with the bounds in Lemma. 2, 3, 4, 5, we have

$$|\mathcal{E}(\widehat{f}_{\lambda,M}) - \inf_{f \in \mathcal{H}} \mathcal{E}(f)|^{1/2} \leq b_1 A + b_1 b_2 B + b_3 \mathcal{C}(\lambda, M) + D,$$

where $b_1 := \|S_M\widehat{C}_{M,\lambda}^{-1}C_{M,\lambda}^{1/2}\|$, $b_2 := \|L_{M,\lambda}^{-1/2}L^{1/2}\|$, $b_3 := \|L_{M,\lambda}^{-1/2}L_\lambda^{1/2}\|$,
$A := \|C_{M,\lambda}^{-1/2}(\widehat{S}_M^* \widehat{y} - S_M^* f_\rho)\|_{\rho X}$, $B := R\kappa^{2r-1}\|C_{M,\lambda}^{-1/2}(C_M - \widehat{C}_M)\|$ and $D := R\lambda^r$. Now note that $b_2 \leq b_3$ for any $\lambda > 0$ since, for any $X, T$, bounded linear operators, with $T$ positive, by multiplying and dividing for $T_\lambda$ the following holds

$$\|XT\| \leq \|XT_\lambda\|\|T_\lambda^{-1}T\|, \quad (24)$$

and $\|T_\lambda^{-1}T\| \leq 1$, for any $\lambda > 0$. Since $A, B, \mathcal{C}(\lambda, M), D$ will contribute to the rates of the bound, while $b_1, b_3$ are responsible for the numerical constants, we are going to bound the excess risk in order to collect $b_1, b_3$ in a multiplicative term as follows

$$|\mathcal{E}(\widehat{f}_{\lambda,M}) - \inf_{f \in \mathcal{H}} \mathcal{E}(f)|^{1/2} \leq \max(1, b_1)\max(1, b_3)(A + B + \mathcal{C}(\lambda, M) + D).$$

Finally, we further simplify $b_1, b_3$, in particular we apply Prop. 8 in the appendix, obtaining $b_3 \leq (1 - \beta_2)^{-1/2}$. For $b_1$ note that,

$$\|S_M\widehat{C}_{M,\lambda}^{-1}C_{M,\lambda}^{1/2}\| \leq \|S_M\widehat{C}_{M,\lambda}^{-1/2}\|\|\widehat{C}_{M,\lambda}^{-1/2}C_{M,\lambda}^{1/2}\| \leq \|\widehat{C}_{M,\lambda}^{-1/2}C_{M,\lambda}^{1/2}\|^2,$$

since, $\|S_M\widehat{C}_{M,\lambda}^{-1/2}\| \leq \|C_{M,\lambda}^{1/2}\widehat{C}_{M,\lambda}^{-1/2}\|$ for the same reasoning in Eq. (24). Then we apply Prop. 8 in the appendix, obtaining $b_1 \leq (1 - \beta_1)^{-1}$. □

In the next subsection, we are going to find probabilistic estimates for terms in the analytic decomposition of the excess risk in Thm. 4.

### A.3 Probabilistic Estimates

In this section we provide bounds in probability for the quantities $\beta, \mathcal{S}, \mathcal{C}$ of Thm. 4 and for the empirical effective dimension. The notation is introduced in Sect. A.1. First, we fix the notation on the random variables used in the rest of the subsection. Recall that $\beta, \mathcal{S}, \mathcal{C}$ are expressed with respect

to the random variables $z := ((x_1, y_1), \ldots, (x_n, y_n))$, and $\omega := (\omega_1, \ldots, \omega_M)$. The associated sample space is $W := Z \times \Omega^M$ and $Z := (X \times \mathbb{R})^n$, with probability measure $\mathbb{P} := \rho^{\otimes n} \otimes \pi^{\otimes M}$. In particular let $Q \subseteq W$ be an event, we denote with $Q|\omega$ the subset of $Z$ associated to the event $Q$ given $\omega$, that is $Q|\omega := \{z \mid (z, \omega) \in Q\}$. By denoting $\rho^{\otimes n}$ with $\mathbb{P}_Z$ and $\pi^{\otimes M}$ by $\mathbb{P}_\Omega$, we recall that

$$\mathbb{P}(Q) = \int_{\Omega^M} \mathbb{P}_Z(Q|\omega) d\mathbb{P}_\Omega(\omega). \tag{25}$$

Moreover, we recall the following basic facts about $\mathcal{F}_\infty(\lambda)$ and $\mathcal{N}(\lambda)$. We can characterize the upper and lower bounds for $\mathcal{F}_\infty(\lambda)$, in particular we have that $\mathcal{F}_\infty(\lambda) \leq \kappa^2 \lambda^{-1}$ when $\psi$ is uniformly bounded by $\kappa$ (see Asm. 3), moreover $\mathcal{F}_\infty(\lambda) \geq \mathcal{N}(\lambda)$ indeed $\mathcal{N}(\lambda)$ is characterized by $\mathcal{N}(\lambda) = \mathbb{E}_\omega \|(L + \lambda I)^{-1/2} \psi_\omega\|_{\rho_X}^2$ (see Eq. 30), so

$$\mathcal{N}(\lambda) = \mathbb{E}_\omega \|(L + \lambda I)^{-1/2} \psi_\omega\|_{\rho_X}^2 \leq \sup_{\omega \in \Omega} \|(L + \lambda I)^{-1/2} \psi_\omega\|_{\rho_X}^2 = \mathcal{F}_\infty(\lambda).$$

### A.3.1 Estimates for $\mathcal{S}(\lambda, M, n)$

The next lemma bounds the first term of $\mathcal{S}(\lambda, M, n)$ and use a similar technique to the one in [10], while Lemma 7 bounds the whole $\mathcal{S}(\lambda, M, n)$. First we need to introduce $\mathcal{N}_M(\lambda)$ that is the effective dimension induced by the kernel $K_M$. For any $\lambda > 0$ define $\mathcal{N}_M(\lambda)$ as follows,

$$\mathcal{N}_M(\lambda) := \mathrm{Tr}((L_M + \lambda I)^{-1} L_M).$$

In Prop. 10 in the appendix, we bound $\mathcal{N}_M(\lambda)$ in terms of the effective dimension $\mathcal{N}(\lambda)$ that is the one associated to the kernel $K$. Prop. 10 refines the result of Prop. 1 of [16], with simpler proof and slightly improved constants.

**Lemma 6.** *Let $\delta \in (0, 1]$, $n, M \in \mathbb{N}$ and $\lambda > 0$. Given $\omega_1, \ldots, \omega_M \in \Omega$, under Assumptions 3, 4, the following holds with probability at least $1 - \delta$*

$$\|C_{M,\lambda}^{-1/2}(\widehat{S}_M^* \widehat{y} - S_M^* f_\rho)\| \leq 2 \left( \frac{B\kappa}{\sqrt{\lambda}n} + \sqrt{\frac{\sigma^2 \mathcal{N}_M(\lambda)}{n}} \right) \log \frac{2}{\delta}.$$

*Proof.* In this proof we bound the quantity under study, by using the Bernstein inequality for sum of zero-mean random vectors (see Prop. 2 in the appendix). Since $\widehat{S}_M^* \widehat{y} = n^{-1} \sum_{i=1}^n \phi_M(x_i) y_i$ (see Def. 2) we have

$$C_{M,\lambda}^{-1/2}(\widehat{S}_M^* \widehat{y} - S_M^* f_\rho) = \frac{1}{n} \sum_{i=1}^n \zeta_i,$$

where $\zeta_1, \ldots, \zeta_n$ are defined as $\zeta_i = z_i - \mu$ with $z_i := C_{M,\lambda}^{-1/2} \phi_M(x_i) y_i$, and $\mu \in \mathbb{R}^M$ defined as $\mu := C_{M,\lambda}^{-1/2} S_M^* f_\rho$, for $1 \leq i \leq n$. Note that $\zeta_1, \ldots, \zeta_n$ are independent and identically distributed random vectors given $\omega_1, \ldots, \omega_M$, since $(x_1, y_1), \ldots, (x_n, y_n)$ are assumed i.i.d. with respect to $\rho$. Moreover note that, by definition of $f_\rho$,

$$\int \phi_M(x) y d\rho(x, y) = \int y d\rho(y|x) d\rho_X(x) = \int \phi_M(x) f_\rho(x) d\rho_X(x) = S_M^* f_\rho.$$

So, by linearity of the expectation the $\zeta_i$,

$$\mathbb{E}z_i = C_{M,\lambda}^{-1/2} \int \phi_M(x) y d\rho(x, y) = C_{M,\lambda}^{-1/2} S_M^* f_\rho = \mu,$$

which implies that $\zeta_i = z_i - \mu$ is a zero-mean random variable for $1 \leq i \leq n$. Let $z$ be another random variable independent and identically distributed as the $z_i$'s. To apply the Bernstein inequality for random vectors, we need to bound their moments. First of all note that for any $p \geq 1$

$$\begin{aligned} \mathbb{E}\|\zeta_i\|^p &= \mathbb{E}\|z_i - \mu\|^p = \mathbb{E}\|z_i - \mathbb{E}z\|^p \\ &\leq \mathbb{E}_{z_i} \mathbb{E}_z \|z_i - z\|^p \leq 2^{p-1} \mathbb{E}_{z_i} \mathbb{E}_z (\|z_i\|^p + \|z\|^p) = 2^p \mathbb{E}_z \|z\|^p. \end{aligned}$$

In particular, by applying Asm. 3 and Asm. 4, we have

$$\mathbb{E}_z \|z\|^p = \int_{X \times \mathbb{R}} \|C_{M,\lambda}^{-1/2} \phi_M(x) y\|^p d\rho(x,y) = \int_X \|C_{M,\lambda}^{-1/2} \phi_M(x)\|^p \int |y|^p d\rho(y|x) \, d\rho_X(x)$$

$$\leq \frac{1}{2} p! \sigma^2 B^{p-2} \int_X \|C_{M,\lambda}^{-1/2} \phi_M(x)\|^p d\rho_X(x)$$

$$\leq \frac{1}{2} p! \sigma^2 B^{p-2} \left( \sup_{x \in X} \|C_{M,\lambda}^{-1/2} \phi_M(x)\|^{p-2} \right) \int_X \|C_{M,\lambda}^{-1/2} \phi_M(x)\|^2 d\rho_X$$

$$= \frac{1}{2} p! \sqrt{J(\lambda)\sigma^2}^2 \left( \frac{B\kappa}{\sqrt{\lambda}} \right)^{p-2}.$$

where $J(\lambda) = \int_X \|C_{M,\lambda}^{-1/2} \phi_M(x)\|^2 d\rho_X(x)$, while $\|C_{M,\lambda}^{-1/2} \phi_M(x)\| \leq \kappa/\sqrt{\lambda}$ a.s. is given by

$$\|C_{M,\lambda}^{-1/2} \phi_M(x)\|^2 \leq \frac{1}{\lambda} \sup_{x \in X} \|\phi_M(x)\|^2 = \frac{1}{\lambda M} \sup_{x \in X} \sum_{i=1}^{M} |\psi_{\omega_i}(x)|^2$$

$$\leq \frac{1}{\lambda M} \sum_{i=1}^{M} \sup_{x \in X} |\psi_{\omega_i}(x)|^2 \leq \frac{1}{\lambda M} \sum_{i=1}^{M} \sup_{\omega \in \Omega, x \in X} |\psi_\omega(x)|^2 \leq \left( \frac{\kappa}{\sqrt{\lambda}} \right)^2,$$

where the last step is due to Asm. 3. Finally, to concentrate the sum of random vectors, we apply Prop. 2. To conclude the proof we need to prove that $J(\lambda) = \mathcal{N}_M(\lambda)$. Note that, by Rem. 8, we have that $L_M = S_M S_M^*$ and $C_M = S_M^* S_M$, so

$$\mathcal{N}_M(\lambda) = \operatorname{Tr} L_M L_{M,\lambda}^{-1} = \operatorname{Tr} S_M^* L_{M,\lambda}^{-1} S_M = \operatorname{Tr} C_M C_{M,\lambda}^{-1},$$

since $L_M = S_M S_M^*$ and $S_M^* L_{M,\lambda}^{-1} S_M = C_M C_{M,\lambda}^{-1}$. By the the ciclicity of the trace and the definition of $C_M$ in Def. 2, we have

$$\operatorname{Tr} C_M C_{M,\lambda}^{-1} = \int_X \operatorname{Tr}(\phi_M(x)\phi_M(x)^\top C_{M,\lambda}^{-1}) d\rho_X(x) = \int_X \|C_{M,\lambda}^{-1/2} \phi_M(x)\|^2 d\rho_X(x) = J(\lambda).$$

$\square$

**Lemma 7** (Bounding $\mathcal{S}(\lambda, M, n)$). *Let $\delta \in (0, 1/3]$, $n \in \mathbb{N}$ and let $\mathcal{S}(\lambda, M, n)$ be as in Thm. 4, point 1. Let $\bar{B} = B + 2R\kappa^{2r}$, $\bar{\sigma} = \sigma + \sqrt{R}\kappa^r$. Under Asm. 3, 4 the following holds with probability at least $1 - 3\delta$*

$$\mathcal{S}(\lambda, m, n) \leq 4 \left( \frac{\bar{B}\kappa}{\sqrt{\lambda}n} + \sqrt{\frac{\bar{\sigma}^2 \mathcal{N}(\lambda)}{n}} \right) \log \frac{2}{\delta}, \tag{26}$$

*when $0 < \lambda < \|L\|$ and $M \geq (4 + 18\mathcal{F}_\infty(\lambda)) \log \frac{12\kappa^2}{\lambda\delta}$.*

*Proof.* Let $0 < \lambda < \|L\|$ and $M \geq (4 + 18\mathcal{F}_\infty(\lambda)) \log \frac{12\kappa^2}{\lambda\delta}$ (the assumption on $\lambda$, $M$ are necessary for the application of Prop. 10). Let $Q \subseteq W$ be the event satisfying Eq. 26. The goal is to prove that the probability associated to the event $Q$ is $\mathbb{P}(Q) \geq 1 - 3\delta$. Since the quantity $\mathcal{S}(\lambda, M, n)$ is defined in Thm. 4 as

$$\mathcal{S}(\lambda, M, n) = \|C_{M,\lambda}^{-1/2} (\widehat{S}_M^* y - S_M^* f_\rho)\| + R\kappa^{2r-1} \|C_{M,\lambda}^{-1/2} (C_M - \widehat{C}_M)\|,$$

we are first going to bound in probability the single terms on the rhs, given $\omega_1, \dots, \omega_M$, then we take the union event, and we use this to prove that $\mathbb{P}(Q) \geq 1 - 3\delta$.

First of all we need to define four other events. Define the event $E_{\boldsymbol{\omega}}^1 \subseteq Z$, as the event satisfying

$$\|C_{M,\lambda}^{-1/2} (\widehat{S}_M^* \widehat{y} - S_M^* f_\rho)\| \leq 2 \left( \frac{B\kappa}{\sqrt{\lambda}n} + \sqrt{\frac{\sigma^2 \mathcal{N}_M(\lambda)}{n}} \right) \log \frac{2}{\delta}. \tag{27}$$

By Lemma 6 we know that $E_{\boldsymbol{\omega}}^1$ holds with probability $\mathbb{P}_Z(E_{\boldsymbol{\omega}}^1) \geq 1 - \delta$ almost everywhere for $\boldsymbol{\omega}$. Then, define the event $E_{\boldsymbol{\omega}}^2 \subset Z$, as the event satisfying

$$\|C_{M,\lambda}^{-1/2} (C_M - \widehat{C}_M)\| \leq \frac{4\kappa^2 \log \frac{2}{\delta}}{\sqrt{\lambda}n} + \sqrt{\frac{4\kappa^2 \mathcal{N}_M(\lambda) \log \frac{2}{\delta}}{n}}. \tag{28}$$

By applying Prop. 5, with $v_i = z_i = \phi_M(x_i)$ for $1 \leq i \leq n$, we have that $E^2_{\boldsymbol{\omega}}$ holds with probability $\mathbb{P}_Z(E^2_{\boldsymbol{\omega}}) \geq 1 - \delta$ almost everywhere for $\boldsymbol{\omega}$. Define $E_{\boldsymbol{\omega}} \subseteq Z$ as the event satisfying

$$\mathcal{S}(\lambda, M, n) \leq 2\left(\frac{B\kappa + 2R\kappa^{2r+1}}{\sqrt{\lambda}\,n} + (\sigma + R\kappa^{2r})\sqrt{\frac{\mathcal{N}_M(\lambda)}{n}}\right)\log\frac{2}{\delta}. \tag{29}$$

Denote with $t$ the right hand side of the equation above and with $s_1, t_1, s_2, t_2$ respectively the lhs and the rhs of Eq. (27), (28). We have that $s_1 \leq t_1$ and $s_2 \leq t_2$ implies $\mathcal{S}(\lambda, M, n) \leq t$, indeed $\mathcal{S}(\lambda, M, n) = s_1 + R\kappa^{2r-1}s_2$ and $t_1 + R\kappa^{2r-1}t_2 \leq t$, since $\log(2/\delta) > 1$. In set terms $(E^1_{\boldsymbol{\omega}} \cap E^2_{\boldsymbol{\omega}}) \subseteq E_{\boldsymbol{\omega}}$, that implies $\mathbb{P}_Z(E_{\boldsymbol{\omega}}) \geq \mathbb{P}_Z(E^1_{\boldsymbol{\omega}} \cap E^2_{\boldsymbol{\omega}})$, in particular

$$\mathbb{P}_Z(E_{\boldsymbol{\omega}}) \geq \mathbb{P}_Z(E^1_{\boldsymbol{\omega}} \cap E^2_{\boldsymbol{\omega}}) \geq \mathbb{P}_Z(E^1_{\boldsymbol{\omega}}) + \mathbb{P}_Z(E^2_{\boldsymbol{\omega}}) - 1 \geq 1 - 2\delta,$$

where we used the fact that for any probability measure $P$ and two events $A, B$, we have $P(A \cap B) = P(A) + P(B) - P(A \cup B) \geq P(A) + P(B) - 1$. The last event that we need to define is $A \subseteq \Omega^M$ satisfying $\mathcal{N}_M(\lambda) \leq 1.5\mathcal{N}(\lambda)$. By Prop. 10, we know that $A$ holds with probability $\mathbb{P}_\Omega(A) \geq 1 - \delta$.

Now we characterize the probability of $Q|\boldsymbol{\omega}$ when $\boldsymbol{\omega} \in A$. Denote with $t_E$ the rhs of Eq.29 defining $E_{\boldsymbol{\omega}}$ and $t_Q$ the rhs of Eq. 26 defining $Q$ (and so $Q|\boldsymbol{\omega}$). When $\boldsymbol{\omega} \in A$, we have that $\mathcal{N}_M(\lambda) \leq 1.5\mathcal{N}(\lambda)$ and so $t_E \leq t_Q$. Then, when $\boldsymbol{\omega} \in A$, we have that $\mathcal{S}(\lambda, M, n) \leq t_E$ implies $\mathcal{S}(\lambda, M, n) \leq t_Q$, that is $E_{\boldsymbol{\omega}} \subseteq Q|\boldsymbol{\omega}$, implying that $\mathbb{P}_Z(E_{\boldsymbol{\omega}}) \leq \mathbb{P}_Z(Q|\boldsymbol{\omega})$. By using the expansion of $\mathbb{P}(Q)$ in Eq. 25 we have

$$\mathbb{P}(Q) = \int_A \mathbb{P}_Z(Q|\boldsymbol{\omega})d\mathbb{P}_\Omega(\boldsymbol{\omega}) + \int_{\Omega^M\setminus A}\mathbb{P}_Z(Q|\boldsymbol{\omega})d\mathbb{P}_\Omega(\boldsymbol{\omega}) \geq \int_A \mathbb{P}_Z(Q|\boldsymbol{\omega})d\mathbb{P}_\Omega(\boldsymbol{\omega})$$

$$\geq \int_A \mathbb{P}_Z(E_\omega)d\mathbb{P}_\Omega(\boldsymbol{\omega}) \geq (1 - 2\delta)\int_A d\mathbb{P}_\Omega(\boldsymbol{\omega}) \geq (1 - 2\delta)(1 - \delta) \geq 1 - 3\delta.$$

$\square$

### A.3.2 Estimates for $\mathcal{C}(\lambda, M)$

**Lemma 8** (Bounding $\mathcal{C}(\lambda, m)$). *Let $\mathcal{C}(\lambda, M)$ as in Thm. 4, point 2. Let $\delta \in (0, 1/2]$ and $\lambda > 0$. Under Asm. 3, following holds with probability at least $1 - 2\delta$*

$$\mathcal{C}(\lambda, m) \leq 4R\kappa^{2r-1}\left(\frac{\sqrt{\lambda\mathcal{F}_\infty(\lambda)}\log\frac{2}{\delta}}{M^r} + \sqrt{\frac{\lambda\mathcal{N}(\lambda)^{2r-1}\mathcal{F}_\infty(\lambda)^{2-2r}\log\frac{2}{\delta}}{M}}\right)t^{1-r},$$

*when $M \geq (4 + 18\mathcal{F}_\infty(\lambda))\log\frac{8\kappa^2}{\lambda\delta}$ and $t := \log\frac{11\kappa^2}{\lambda}$.*

*Proof.* We now study $\mathcal{C}(\lambda, M)$ that is

$$\mathcal{C}(\lambda, M) = R\lambda^{1/2}\|L_\lambda^{-1/2}(L - L_M)\|^{2r-1}\|L_\lambda^{-1/2}(L - L_M)L_\lambda^{-1/2}\|^{2-2r}.$$

We are going to bound the two terms in probability, via Prop. 5 and Prop. 6. First of all, we recall that $\mathcal{F}_\infty(\lambda) := \sup_{\omega\in\Omega}\|L_\lambda^{-1/2}\psi_\omega\|^2_{\rho_X}$ and that $\mathcal{N}(\lambda) = \mathbb{E}\|L_\lambda^{-1/2}\psi_\omega\|^2_{\rho_X}$, indeed by Lemma. 1, characterizing $L$ in terms of $\psi_\omega$, the linearity and the ciclicity of the trace, we have,

$$\mathcal{N}(\lambda) := \mathrm{Tr}(L_\lambda^{-1}L) = \mathrm{Tr}\left(L_\lambda^{-1}\int\psi_\omega\otimes\psi_\omega d\pi(\omega)\right) = \int\mathrm{Tr}(L_\lambda^{-1}(\psi_\omega\otimes\psi_\omega))d\pi(\omega) \tag{30}$$

$$= \int\langle\psi_\omega, L_\lambda^{-1}\psi_\omega\rangle_{\rho_X}d\pi(\omega) = \int\|L_\lambda^{-1/2}\psi_\omega\|^2_{\rho_X}d\pi(\omega), \tag{31}$$

where the last steps are due to the identity $\mathrm{Tr}(A(v\otimes v)) = \langle v, Av\rangle = \|A^{1/2}v\|^2$ valid for any vector $v$ and any bounded self-adjoint positive operator $A$ on a Hilbert space.

Define $A \subseteq \Omega^M$ the event satisfying

$$\|L_\lambda^{-1/2}(L - L_M)\| \leq \frac{4\sqrt{\mathcal{F}_\infty(\lambda)\kappa^2}\log\frac{2}{\delta}}{M} + \sqrt{\frac{4\kappa^2\mathcal{N}(\lambda)\log\frac{2}{\delta}}{M}}.$$

By the fact that $\|\cdot\| \le \|\cdot\|_{HS}$ and by Prop. 5, with $v_i = L_\lambda^{-1/2}\psi_{\omega_i}$ and $z_i = \psi_{\omega_i}$ for $i \in \{1, \dots, M\}$ and $Q = T = L$, $T_n = L_M$, we know that the event $A$ has probability $\mathbb{P}_\Omega(A) \ge 1 - \delta$.

Define $B \subseteq \Omega^M$ the event satisfying

$$\|L_\lambda^{-1/2}(L - L_M)L_\lambda^{-1/2}\| \le \frac{2\eta(1 + \mathcal{F}_\infty(\lambda))}{3M} + \sqrt{\frac{2\eta\mathcal{F}_\infty(\lambda)}{M}}, \tag{32}$$

with $\eta := \log \frac{8\kappa^2}{\lambda\delta}$. By Prop. 6, with $Q = L$ and $v_i = \psi_{\omega_i}$ for $i \in \{1, \dots, M\}$, we know that $B$ has probability $\mathbb{P}_\Omega(B) \ge 1 - \delta$.

Now set $E = A \cap B$. When $E$ holds and under the assumption that $M \ge (4 + 18\mathcal{F}_\infty(\lambda))\log \frac{8\kappa^2}{\lambda\delta}$, we have that the right hand side of Eq. (32) is smaller than $\sqrt{\frac{4\eta\mathcal{F}_\infty(\lambda)}{M}}$, and $\eta = \log \frac{2}{\delta} + \log \frac{4\kappa^2}{\lambda}$, so

$$\mathcal{C}(\lambda, M) \le R\lambda^{1/2}\left(\frac{4\sqrt{\mathcal{F}_\infty(\lambda)\kappa^2}\log \frac{2}{\delta}}{M} + \sqrt{\frac{4\kappa^2\mathcal{N}(\lambda)\log \frac{2}{\delta}}{M}}\right)^{2r-1}\sqrt{\frac{4\eta\mathcal{F}_\infty(\lambda)}{M}}^{2-2r} \tag{33}$$

$$\le 4R\kappa^{2r-1}\left(\frac{\sqrt{\lambda\mathcal{F}_\infty(\lambda)}(\log \frac{2}{\delta})^r}{M^r} + \sqrt{\frac{\lambda\mathcal{N}(\lambda)^{2r-1}\mathcal{F}_\infty(\lambda)^{2-2r}\log \frac{2}{\delta}}{M}}\right)\left(1 + \frac{\log \frac{4\kappa^2}{\lambda}}{\log \frac{2}{\delta}}\right)^{1-r}. \tag{34}$$

The event $E$ holds with probability $\mathbb{P}_\Omega(E) \ge \mathbb{P}_\Omega(A) + \mathbb{P}_\Omega(B) - 1 \ge 1 - 2\delta$. We recall that since $E$ does not depend on $z$, the probability of $E$ in $W$ is given by the canonical extension $E' = Z \times E$ whose probability is again $\mathbb{P}(E') = \mathbb{P}_\Omega(E) \ge 1 - 2\delta$. Finally, we further upper bound in Eq.(33) the term $(\log \frac{2}{\delta})^r$ with $\log \frac{2}{\delta}$ since $\log \frac{2}{\delta} > 1$, $r \in [1/2, 1]$, and $1 + (\log \frac{4\kappa^2}{\lambda})/(\log \frac{2}{\delta})$ with $\log \frac{11\kappa^2}{\lambda}$, for the same reasons and the fact that $1 + \log 4 = \log 4e \le \log 11$. $\qquad\square$

### A.3.3  Estimates for $\beta$

**Lemma 9** (Bounding the norm of $C_M$). *Let $\delta \in (0, 1]$. Under Asm. 3, the following holds with probability at least $1 - \delta$,*

$$\|C_M\| \ge \frac{3}{4}\|L\|,$$

*when $M \ge 32\left(\frac{\kappa^2}{\|L\|} + \kappa^2\right)\log \frac{2}{\delta}$.*

*Proof.* Define the event $A \in \Omega^M$ as the one satisfying

$$\|L - L_M\|_{HS} \le \frac{4\kappa^2}{M}\log \frac{2}{\delta} - \sqrt{\frac{2\kappa^2}{M}\log \frac{2}{\delta}}.$$

Note that when $\omega \in A$ and $M \ge 32\left(\frac{\kappa^2}{\|L\|} + \kappa^2\right)\log \frac{2}{\delta}$, we have $\|L - L_M\|_{HS} \le \frac{1}{4}\|L\|$ and, by using the characterization of $C_M, L_M$ in Rem. 8 and the fact that $\|\cdot\|_{HS} \ge \|\cdot\|$, we have

$$\|C_M\| = \|S_M^* S_M\| = \|S_M S_M^*\| = \|L_M\| \ge |\|L\| - \|L - L_M\||$$

$$\ge \|L\| - \|L - L_M\| \ge \|L\| - \|L - L_M\|_{HS} \ge \|L\| - \frac{1}{4}\|L\| \ge \frac{3}{4}\|L\|.$$

Now we find a lower bound for the probability of $A$. Let $\zeta_i = L - \psi_{\omega_i} \otimes \psi_{\omega_i}$ be a random operator with $\omega_i$ independently and identically distributed w.r.t $\pi$ and $i \in \{1, \dots, M\}$. We have that $L - L_M = \frac{1}{M}\sum_{i=1}^M \zeta_i$ and $\mathbb{E}\zeta_i = 0$, by the characterization of $L$ in Lemma 1. Denote with $\mathcal{L}$ the Hilbert space of Hilbert-Schmidt operators on $L^2(X, \rho_X)$. Now note that, since it is trace class, $\zeta_i$ is a random vector belonging to $\mathcal{L}$, so we can apply Prop. 2, with $T = \sup_{\omega \in \Omega}\|L - \psi_\omega \otimes \psi_\omega\|_{HS} \le 2\kappa^2$ and $S = \mathbb{E}\|\zeta_1\|_{HS}^2 \le \kappa^2$, obtaining that $\mathbb{P}_\Omega(A) \ge 1 - \delta$. $\qquad\square$

**Lemma 10** (Bounding $\beta$). *Let $\delta \in (0, 1/3]$, and $\beta$ be as in Thm. 4, point 3. Under Asm. 3, the following holds with probability at least $1 - 3\delta$,*

$$\beta < 2,$$

*when* $0 < \lambda \leq \frac{3}{4}\|L\|$, *and*

$$n \geq 18\left(2 + \frac{\kappa}{\lambda}\right)\log\frac{4\kappa^2}{\lambda\delta}, \qquad M \geq 18\left(2 + \mathcal{F}_\infty(\lambda)\right)\log\frac{4\kappa^2}{\lambda\delta} \vee 32\left(\frac{\kappa^2}{\|L\|} + \kappa^2\right)\log\frac{2}{\delta}.$$

*Proof.* Let $\beta, \beta_1, \beta_2$ be defined as in Thm. 4, point 3. To bound $\beta$ in probability, we first bound $\beta_1$ and $\beta_2$ in probability, and then, under the intersection of the events, we control $\beta$. First of all, denote with $(a)$ the condition on $\lambda$, with $(b)$ the condition on $n$ and with $(c.1)$ the condition $M \geq 32\left(\frac{\kappa^2}{\|L\|} + \kappa^2\right)\log\frac{2}{\delta}$, while with $(c.2)$ the condition $M \geq 18\left(2 + \mathcal{F}_\infty(\lambda)\right)\log\frac{4\kappa^2}{\lambda\delta}$. Define the event $E \subseteq W$ as the one satisfying $\beta_1 \leq \frac{1}{3}$. To bound the probability of $E$ we need an auxiliary event $A \in \Omega^M$ that is the one satisfying $\frac{3}{4}\|L\| \leq \|C_M\|$. The specific choice of $A$ will be made clear later. We have

$$\mathbb{P}(E) = \int_A \mathbb{P}_Z(E|\boldsymbol{\omega})d\mathbb{P}_\Omega(\boldsymbol{\omega}) + \int_{\Omega^M\setminus A} \mathbb{P}_Z(E|\boldsymbol{\omega})d\mathbb{P}_\Omega(\boldsymbol{\omega}) \geq \int_A \mathbb{P}_Z(E|\boldsymbol{\omega})d\mathbb{P}_\Omega(\boldsymbol{\omega}).$$

By Prop. 6 and Rem. 10 point 3, we know that $\mathbb{P}_Z(E|\boldsymbol{\omega}) \geq 1 - \delta$, for any $\boldsymbol{\omega}$, when $\lambda \leq \|C_M\|$ and condition $(b)$ hold. Note that, when $\boldsymbol{\omega}$ is in $A$ then $\frac{3}{4}\|L\| \leq \|C_M\|$ and so the condition $\lambda \leq \|C_M\|$ is always satisfied by assuming $(a)$. Then under $(a), (b)$, we have $\mathbb{P}_Z(E|\boldsymbol{\omega}) \geq 1 - \delta$ when $\boldsymbol{\omega} \in A$, and so under the same conditions

$$\mathbb{P}(E) \geq \int_A \mathbb{P}_Z(E|\boldsymbol{\omega})d\mathbb{P}_\Omega(\boldsymbol{\omega}) \geq (1 - \delta)\int_A d\mathbb{P}_\Omega(\boldsymbol{\omega}) = (1 - \delta)\mathbb{P}_\Omega(A).$$

By Lemma 9, $\mathbb{P}_\Omega(A) \geq 1 - \delta$, when $(c.1)$ holds, so $\mathbb{P}(E) \geq (1 - \delta)\mathbb{P}_\Omega(A) \geq (1 - \delta)^2 \geq 1 - 2\delta$ when $(a), (b), (c.1)$ hold.

Define $D_0 \subseteq \Omega^M$ as the event satisfying $\beta_2 \leq \frac{1}{3}$. By Prop. 6 and Rem. 10 point 3, we know that $\mathbb{P}_\Omega(D_0) \geq 1 - \delta$, when the condition $(c.2)$ and $(a)$ hold. So the event $D := Z \times D_0$ has probability $\mathbb{P}(D) = \mathbb{P}_\Omega(D_0) = 1 - \delta$, when $(c.2)$ holds. Finally, note that under the conditions $(a), (b), (c.1), (c.2)$, when the event $D \cap E$ hold, we have $\beta \leq (2/3)^{-3/2} < 2$. The probability of $D \cap E$ under the conditions $(a), (b), (c.1), (c.2)$ has probability

$$\mathbb{P}(D \cap E) = \mathbb{P}(D) + \mathbb{P}(E) - \mathbb{P}(D \cup E) \geq \mathbb{P}(D) + \mathbb{P}(E) - 1 \geq 1 - 3\delta.$$

$\square$

## A.4 Proof of the Main Result

Here we prove Thm. 1, 2, 3 that are the main results of the paper. In particular the following Thm. 5 is a general version of the three theorem above, without the need of Assumptions 5, 2 and valid for a wide range of $\lambda, M$. In Thm. 6, we specialize the result of Thm. 5, selecting $M$ in terms of $\lambda$ such that the upper bound of the excess risk depends only on $\lambda$ and is proportional to the same upper bound for kernel ridge regression that leads to optimal generalization bounds. Note that Thm. 6 is again independent of Asm. 5, 2. Finally, Thm. 7 is obtained by Thm. 6, by adding Asm. 5, 2 and has all the constant explicit. Then Thm. 1 is a specification of Thm. 7, for the simple scenario where it is only required the existence of $f_\mathcal{H}$ (that is Asm. 5 satisfied with $\gamma = 1$, Asm. 6 satisfied with $r = 1/2$ and Asm. 2 with $\alpha = 1$, see discussion after the introduction of the assumptions). Thm. 2 is a specification of Thm. 7 for the fast rates (Asm. 2 is satisfied with $\alpha = 1$), while Thm. 3 is a simplified version of Thm. 7 where the constants have been hidden.

**Theorem 5** (Generalization Bound for RF-KRLS). *Let $\delta \in (0, 1]$. Let $\widehat{f}_{\lambda, M}$ be as in Eq. (7). Under Asm. 3, 6, 4 when $0 < \lambda \leq \frac{3}{4}\|L\|$ and*

$$n \geq 18\left(2 + \frac{\kappa^2}{\lambda}\right)\log\frac{36\kappa^2}{\lambda\delta},$$

$$M \geq 18\left(q_0 + \mathcal{F}_\infty(\lambda)\right)\log\frac{108\kappa^2}{\lambda\delta},$$

*with $q_0 = 2(2 + \frac{\kappa}{\|L\|} + \kappa^2)$, then the following holds with probability at least $1 - \delta$,*

$$\sqrt{\mathcal{E}(\widehat{f}_{\lambda, M}) - \inf_{f \in \mathcal{H}}\mathcal{E}(f)} \leq 2\left(\frac{4\bar{B}\kappa}{\sqrt{\lambda}n} + \sqrt{\frac{16\bar{\sigma}^2\mathcal{N}(\lambda)}{n}} + \mathfrak{C}(\lambda, M) + R\lambda^r\right)\log\frac{18}{\delta}. \qquad (35)$$

where $\bar{B} = B + 2R\kappa$, $\bar{\sigma} = \sigma + 2\sqrt{R}\kappa$,

$$\mathfrak{C}(\lambda, M) := R\kappa^{2r-1}\left(\frac{\sqrt{\lambda\mathcal{F}_\infty(\lambda)}}{M^r} + \sqrt{\frac{\lambda\mathcal{N}(\lambda)^{2r-1}\mathcal{F}_\infty(\lambda)^{2-2r}}{M}}\right)t^{1-r}. \tag{36}$$

and $t := \log\frac{11\kappa^2}{\lambda}$.

*Proof.* Under Asm. 3, the existence of $f_\mathcal{H}$ in Asm. 4 and Asm. 6, plus Rem. 6, we have the following analytical decomposition of the excess risk, by Thm, 4

$$|\mathcal{E}(\widehat{f}_{\lambda,M}) - \inf_{f\in\mathcal{H}}\mathcal{E}(f)|^{1/2} \le \beta\left(\mathcal{S}(\lambda, M, n) + \mathcal{C}(\lambda, M) + R\lambda^r\right), \tag{37}$$

where the quantities $\beta$, $\mathcal{C}(\lambda, M)$ and $\mathcal{S}(\lambda, M, n)$ are defined in the statement of Thm. 4. Under the same assumptions, Lemma 7, 8 and 10 are devoted to bound in probability the three quantities, with the help of the concentration inequalities recalled in Section B, plus some auxiliary results in Section D, of the appendixes.

Let $\tau := \delta/9$. Define the event $D \subseteq W$ as the one satisfying $\beta < 2$ (see Subsection A.3 for the definition of the sample space $W$ for learning with random features, and the associated probability measure $\mathbb{P}$). By Lemma 10 we know that the event $D$ has probability $\mathbb{P}(D) \ge 1 - 3\tau$, when the following conditions hold

$$(d_1)\ \ 0 \le \lambda \le \frac{3}{4}\|L\|, \qquad (d_2)\ \ n \ge 18\,(2 + \kappa/\lambda)\log\frac{4\kappa^2}{\lambda\tau},$$

$$(d_3)\ \ M \ge 18\,(2 + \mathcal{F}_\infty(\lambda))\log\frac{4\kappa^2}{\lambda\tau} \ \vee\ 32\left(\frac{\kappa^2}{\|L\|} + \kappa^2\right)\log\frac{2}{\tau}.$$

Define the event $E \subseteq W$ as the one satisfying

$$\mathcal{S}(\lambda, M, n) \le 4\left(\frac{\bar{B}\kappa}{\sqrt{\lambda}n} + \sqrt{\frac{\bar{\sigma}^2\mathcal{N}(\lambda)}{n}}\right)\log\frac{2}{\tau}. \tag{38}$$

By Lemma 7 we know that the probability of $E$ is $\mathbb{P}(E) \ge 1 - 4\tau$, when the following conditions hold

$$(e_1)\ \ 0 < \lambda < \|L\|, \qquad (e_2)\ \ M \ge (4 + 18\mathcal{F}_\infty(\lambda))\log\frac{12\kappa^2}{\lambda\tau}.$$

Define the event $G \subseteq W$ as the one satisfying

$$\mathcal{C}(\lambda, M) \le \mathfrak{C}(\lambda, M),$$

By Lemma 8 we know that $G$ holds with probability $\mathbb{P}(G) \ge 1 - 2\tau$, when the $(d_1)$ and the following condition holds

$$(g_1)\ \ M \ge (4 + 18\mathcal{F}_\infty(\lambda))\log\frac{8\kappa^2}{\lambda\tau}.$$

Finally Eq. (35) is obtained from Eq. (37), by bounding $\beta$ with 2, $\mathcal{S}(\lambda, M, n)$ with Eq. (38) and $\mathcal{C}(\lambda, M)$ by $\mathfrak{C}(\lambda, M)$. So by definition, Eq. (35) holds under the event $D \cap E \cap G$ and the conditions $(d_1), (e_1)$ on $\lambda$, $(d_2)$ on $n$ and $(d_3), (e_2), (g_1)$ on $M$. The event $D \cap E \cap G$ has probability

$$\begin{aligned}
\mathbb{P}(D \cap E \cap G) &= \mathbb{P}(W \setminus ((W \setminus D) \cup (W \setminus E) \cup (W \setminus G))) \\
&\ge 1 - [\,(1 - \mathbb{P}(D)) + (1 - \mathbb{P}(E)) + (1 - \mathbb{P}(G))\,] \\
&= \mathbb{P}(D) + \mathbb{P}(E) + \mathbb{P}(G) - 2 \ \ge \ 1 - 9\tau.
\end{aligned}$$

Finally note that the conditions on $\lambda, n, M$ in the statement of this theorem imply, respectively, conditions $(d_1), (e_1)$ on $\lambda$, $(d_2)$ on $n$, and $(d_3), (e_2), (g_1)$ on $M$. $\qquad\square$

**Theorem 6** (Generalization Bound for RF-KRLS). *Let $\delta \in (0, 1]$. Let $\widehat{f}_{\lambda,M}$ be as in Eq. (7). Under Asm. 3, 6, 4, when $0 < \lambda \le \frac{3}{4}\|L\|$ and*

$$n \ge 18\left(2 + \frac{\kappa^2}{\lambda}\right)\log\frac{36\kappa^2}{\lambda\delta},$$

$$M \ge 4\kappa^2\left(\frac{\mathcal{N}(\lambda)}{\lambda}\right)^{2r-1}\left(\mathcal{F}_\infty(\lambda)\log\frac{11\kappa^2}{\lambda}\right)^{2-2r} \ \vee\ 18\,(q_0 + \mathcal{F}_\infty(\lambda))\log\frac{108\kappa^2}{\lambda\delta},$$

*with $q_0 = 2(2 + \frac{\kappa}{\|L\|} + \kappa^2)$, then the following holds with probability at least $1 - \delta$,*

$$\sqrt{\mathcal{E}(\widehat{f}_{\lambda,M}) - \inf_{f \in \mathcal{H}} \mathcal{E}(f)} \leq 8 \left( \frac{\bar{B}\kappa}{\sqrt{\lambda}n} + \sqrt{\frac{\bar{\sigma}^2 \mathcal{N}(\lambda)}{n}} + R\lambda^r \right) \log \frac{18}{\delta}. \tag{39}$$

*Here $\bar{B} = B + 2R\kappa$, $\bar{\sigma} = \sigma + 2\sqrt{R}\kappa$.*

*Proof.* First we apply Thm. 5, then we add a condition on $M$ with respect to $\lambda$ such that we can bound $\mathfrak{C}(\lambda, M)$ with $R\lambda^r$. The condition we will consider is the following

$$(g_2) \quad M \geq 4\kappa^2 \lambda^{1-2v} \mathcal{N}(\lambda)^{2v-1} \mathcal{F}_\infty(\lambda)^{2-2v} t^{2-2r}.$$

Indeed lower bounding with $(g_2)$ the occurrences of $M$ in $\mathfrak{C}(\lambda, M)$, we have

$$\mathfrak{C}(\lambda, M) \leq R\kappa^{2r-1} \left( \sqrt{\frac{\lambda^{1+4r^2-2r} \mathcal{F}_\infty(\lambda)^{1+4r^2-4r}}{4^{2r}\kappa^{4r-2}\mathcal{N}(\lambda)^{4r^2-2r}t^{6r-4r^2-2}}} + \sqrt{\frac{\lambda^{2r}}{4\kappa^2}} \right)$$

$$\leq R \left( \sqrt{\frac{\lambda^{2r}}{4^{2r}\kappa^{12r-8r^2-4}\mathcal{N}(\lambda)^{4r^2-2r}t^{6r-4r^2-2}}} + \sqrt{\frac{\lambda^{2r}}{4\kappa^{4-4r}}} \right) \leq R\lambda^r,$$

where the second step is due to $\mathcal{F}_\infty(\lambda) \leq \kappa^2/\lambda$ and $1 + 4r^2 - 4r \geq 0$ for $r \geq 1/2$, while the last step is due to the following three facts. First, that $4^{2r}\mathcal{N}(\lambda)^{4r^2-2r} \geq 4$, since $4r^2 - 2r \geq 0$ on $r \in [1/2, 1]$ and, by denoting with $(\lambda_i(L))_{i\geq 1}$ the eigenvalues of $L$, with $\|L\| := \lambda_1(L) \geq \lambda_2(L) \geq \cdots \geq 0$, and recalling that $0 \leq \lambda \leq \frac{3}{4}\|L\|$, we have

$$\mathcal{N}(\lambda) := \mathrm{Tr}(LL_\lambda^{-1}) = \sum_{i \geq 1} \frac{\lambda_i(L)}{\lambda_i(L) + \lambda} \geq \frac{\lambda_1(L)}{\lambda_1(L) + \lambda} := \frac{\|L\|}{\|L\| + \lambda} > 1/2.$$

Second, that $t^{6r-4r^2-2} \geq 1$, since $6r - 4r^2 - 2 \geq 0$ on $r \in [1/2, 1]$ and $t \geq 1$, since $0 \leq \lambda \leq \frac{3}{4}\|L\| \leq \frac{3}{4}\kappa^2$. Third, that $\kappa^{12r-8r^2-4} \geq 1$ and $\kappa^{4-4r} \geq 1$, since $12r - 8r^2 - 4 \geq 0$ and $4 - 4r \geq 0$ on $r \in [1/2, 1], \kappa \geq 1$. $\qquad \square$

The following theorem is a specialization of the previous one, under 5, 2 and an explicit relation of $\lambda$ with respect to $n$.

**Theorem 7.** *Let $\delta \in (0, 1]$. Under Asm. 3 and 5, 6, 2, 4, let $p := (2r + \gamma - 1)^{-1}$, and*

$$n \geq (2/\|L\|)^{\frac{p+1}{p}} \vee \left( 264\kappa^2 p \log(556\kappa^2 \delta^{-1} \sqrt{p\kappa^2}) \right)^{1+p}$$

$$\lambda_n = n^{-\frac{1}{2r+\gamma}},$$

$$M_n \geq c_0 n^{\frac{\alpha+(2r-1)(1+\gamma-\alpha)}{2r+\gamma}} \log \frac{108\kappa^2}{\lambda\delta},$$

*with $c_0 = 9(3 + 4\kappa^2 + \frac{4\kappa^2}{\|L\|} + \frac{\kappa^2}{4}Q^{2r-1}F^{2-2r})$, then the following holds with probability at least $1 - \delta$,*

$$\mathcal{E}(\widehat{f}_{\lambda_n, M_n}) - \inf_{f \in \mathcal{H}} \mathcal{E}(f) \leq c_1 \log^2 \frac{18}{\delta} n^{-\frac{2r}{2r+\gamma}}, \tag{40}$$

*and $c_1 = 64(\bar{B}\kappa + \bar{\sigma}Q + R)^2$.*

*Proof.* Let $\lambda = n^{-\frac{1}{2r+\gamma}}$ in Thm. 6 and substitute $\mathcal{F}_\infty(\lambda)$ and $\mathcal{N}(\lambda)$ by their bounds given in Asm. 5, 2. Note that to guarantee that $n$ satisfies the associated constraint with respect to $\lambda$, in Thm. 6, and that $\lambda$ is in $(0, \frac{3}{4}\|L\|]$ we need that $n \geq (\frac{4}{3\|L\|})^{\frac{p+1}{p}}$ and

$$n \geq \left( 264\kappa^2 p \log \frac{556\kappa^2 \sqrt{p\kappa^2}}{\delta} \right)^{1+p},$$

with $p = \frac{1}{2r+\gamma-1}$. $\qquad \square$

Note that the theorems in Sect 3 are corollaries of the theorem above. For the sake of readability, in contrast to Thm. 7, the results in Thm. 1, 2, 3 are expressed with respect to $\tau := \log \frac{1}{\delta}$. Moreover in the statement of Thm. 1, 2, 3 the constants and the logarithmic terms are omitted. They can be recovered by plugging the coefficients detailed in the following proofs in the statement of Thm. 7.

**Proof of Theorem 1.** This is an application of Thm. 7, with minimum number of assumptions. Indeed the existence of $f_{\mathcal{H}}$ and the fact that $|y| \leq b$ a. s. satisfies Asm. 4 with $\sigma = B = 2b$. The fact that $X$ is a Polish space and that $\psi$ is bounded continuous satisfy Asm. 3, and so the kernel is bounded by $\kappa^2$. Since the kernel is bounded, we have that Asm. 5 is always satisfied with $\gamma = 1, Q = \kappa$; Asm. 6 is always satisfied with $r = 1/2, R = 1 \vee \|f_{\mathcal{H}}\|_{\mathcal{H}}$; Asm. 2 is always satisfied with $\alpha = 1, F = \kappa^2$.

In particular we have the following constants $n_0 := 4\|L\|^{-2} \vee \left(264\kappa^2 \log \frac{556\kappa^3}{\delta}\right)^2$,

$$c_0 := 9\left(3 + 4\kappa^2 + \frac{4\kappa^2}{\|L\|} + \kappa^4/4\right), \qquad c_1 := 8(\bar{B}\kappa + \bar{\sigma}\kappa + 1 \vee \|f_{\mathcal{H}}\|_{\mathcal{H}}),$$

with $\bar{B} := 2b + 2\kappa(1 \vee \|f_{\mathcal{H}}\|_{\mathcal{H}})$ and $\bar{\sigma} := 2b + 2\kappa\sqrt{1 \vee \|f_{\mathcal{H}}\|_{\mathcal{H}}}$. $\qquad\square$

**Corollary 1.** *Under the same assumptions of Thm. 1, if $n \geq \|L\|^{-2} \vee \left(1056 \log \frac{1056\sqrt{278\kappa^5 b}}{\sqrt{c_1}}\right)^2$ and $\lambda_n = n^{-1/2}$, then a number of random features $M_n$ equal to*

$$M_n = 2c_0 \sqrt{n} \log (c_2 n),$$

*is enough to guarantee that*

$$\mathbb{E} \ \mathcal{E}(\widehat{f}_{\lambda_n, M_n}) - \mathcal{E}(f_{\mathcal{H}}) \leq \frac{40c_1}{\sqrt{n}}.$$

*In particular the constants $c_0, c_1$ are as in Thm. 1 and $c_2 = \frac{8\kappa^2\sqrt{b}}{\sqrt{c_1}}$.*

*Proof.* In the rest we will denote $\mathcal{E}(\widehat{f}_{\lambda_n, M_n}) - \mathcal{E}(f_{\mathcal{H}})$, with $\mathcal{R}(\widehat{f}_{\lambda_n, M_n})$ and will use the notation of Sect. A.3. Fix $\delta_0 = \frac{2c_1}{\kappa^2 b} n^{-1}$. Denote with $E$, the event satisfying $\mathcal{R}(\widehat{f}_{\lambda_n, M_n}) > t_0$, with $t_0 := c_1 \log^2 \frac{18}{\delta_0} n^{-1/2}$.

First, note that $M_n \geq 2c_0\sqrt{n}\log\left(\frac{8\kappa^2\sqrt{b}}{\sqrt{c_1}}n\right)$, satisfies $M_n \geq c_0 \sqrt{n}\log \frac{108\kappa^2\sqrt{n}}{\delta_0}$ and any $n \geq \|L\|^{-2} \vee \left(1056 \log \frac{1056\sqrt{278\kappa^5 b}}{\sqrt{c_1}}\right)^2$ satisfies $n \geq n_0(\delta_0)$ with $n_0(\delta)$ as in Thm. 1. So, we can apply Thm. 1, from which we know that $E$ holds with probability smaller than $\delta_0$.

Second, by Rem. 6 and Rem. 9, we have that

$$\mathcal{R}(\widehat{f}_{\lambda_n, M_n}) = \|S_M \widehat{C}_{M,\lambda}^{-1} \widehat{S}_M^* \widehat{y} - Pf_\rho\|_{\rho_X} \tag{41}$$

$$\leq \|S_M\| \|\widehat{C}_{M,\lambda}^{-1}\| \|\widehat{S}_M^*\| \|\widehat{y}\|_{\mathbb{R}^n} + \|P\| \|f_\rho\|_{\rho_X} \tag{42}$$

$$\leq \frac{\kappa^2 b}{\lambda} + b \leq \frac{2\kappa^2 b}{\lambda} =: R_0, \tag{43}$$

where we used the fact that $\|S_M\|, \|\widehat{S}_M^*\| \leq \kappa$ (see Def. 2), that $\|\widehat{C}_{M,\lambda}^{-1}\| \leq \lambda^{-1}$, that $\|\widehat{y}\|^2 = \frac{1}{n}\sum_{y_i}^2$, that $f_\rho(x) = \mathbb{E}[y|x]$, that the $y$'s are bounded in $[-b, b]$, and the fact that $\kappa^2/\lambda \geq 1$, by definition of $\kappa, \lambda$.

Now, by denoting with $\mathbf{1}_E$ the indicator function for $E$, we have

$$\mathbb{E} \ \mathcal{R}(\widehat{f}_{\lambda_n, M_n}) = \mathbb{E} \ \mathbf{1}_E \mathcal{R}(\widehat{f}_{\lambda_n, M_n}) \ + \ \mathbb{E} \ \mathbf{1}_{Z\backslash E}\mathcal{R}(\widehat{f}_{\lambda_n, M_n}).$$

In particular

$$\mathbb{E} \ \mathbf{1}_E \mathcal{R}(\widehat{f}_{\lambda_n, M_n}) \leq R_0 \ \mathbb{E} \ \mathbf{1}_E = R_0 \mathbb{P}(E) \leq R_0 \delta_0.$$

For the second term we have

$$\mathbb{E} \ \mathbf{1}_{Z\backslash E}\mathcal{R}(\widehat{f}_{\lambda_n, M_n}) \quad = \quad \mathbb{E} \ \mathbf{1}_{\{\mathcal{R}(\widehat{f}_{\lambda_n, M_n}) \leq t_0\}} \mathcal{R}(\widehat{f}_{\lambda_n, M_n}) \quad = \quad \int_0^{t_0} \mathbb{P}(\mathcal{R}(\widehat{f}_{\lambda_n, M_n}) > t)dt.$$

By changing variable, in the integral above, via $t = \frac{c_1}{\sqrt{n}} \log^2 \frac{18}{\delta}$, and using the fact that $\mathbb{P}(\mathcal{R}(\widehat{f}_{\lambda_n, M_n}) > \frac{c_1}{\sqrt{n}} \log^2 \frac{18}{\delta}) \leq \delta$, we have

$$
\begin{aligned}
\int_0^{t_0} \mathbb{P}(\mathcal{R}(\widehat{f}_{\lambda_n, M_n}) > t) dt &= \frac{2c_1}{\sqrt{n}} \int_{\delta_0}^{18} \frac{\log \frac{18}{\delta}}{\delta} \mathbb{P}\left(\mathcal{R}(\widehat{f}_{\lambda_n, M_n}) > \frac{c_1}{\sqrt{n}} \log^2 \frac{18}{\delta}\right) d\delta \\
&\leq \frac{2c_1}{\sqrt{n}} \int_{\delta_0}^{18} \log \frac{18}{\delta} d\delta \\
&= \frac{2c_1}{\sqrt{n}} \left(18 - \delta_0 \left(1 + \log \frac{18}{\delta_0}\right)\right) \quad \leq \quad \frac{36c_1}{\sqrt{n}}.
\end{aligned}
$$

So finally we have

$$
\mathbb{E}\, \mathcal{R}(\widehat{f}_{\lambda_n, M_n}) \leq R_0 \delta_0 + \frac{36c_1}{\sqrt{n}} = \frac{40c_1}{\sqrt{n}}.
$$

$\square$

**Proof of Theorem 2.** This is an application of Thm. 7, where assumption Asm. 2 is satisfied with $F = \kappa^2$ and $\alpha = 1$. Indeed the existence of $f_{\mathcal{H}}$ and the fact that $|y| \leq b$ a. s. satisfies Asm. 4 with $\sigma = B = 2b$. The fact that $X$ is a Polish space and that $\psi$ is bounded continuous satisfy, Asm. 3, and so the kernel is bounded by $\kappa^2$. Since the kernel is bounded, we have that Asm. 2 is always satisfied with $\alpha = 1, F = \kappa^2$. Asm. 4 and Asm. 6 are directly satisfied by Asm. 1. In particular we obtain

$$
n_0 := (2/\|L\|)^{\frac{p+1}{p}} \vee \left(264\kappa^2 p \, \log(556\kappa^2 \delta^{-1} \sqrt{p\kappa^2})\right)^{1+p},
$$

$$
c_0 := 9\left(3 + 4\kappa^2 + \frac{4\kappa^2}{\|L\|} + \frac{\kappa^{4-2r}}{4} Q^{2r-1}\right), \quad c_1 := 64\left(\bar{B}\kappa + \bar{\sigma}Q + R\right)^2,
$$

with $\bar{B} := 2b + 2\kappa R$ and $\bar{\sigma} := 2b + 2\kappa\sqrt{R}$. $\square$

**Proof of Example 2.** By definition of $\psi_s, \pi_s$ we have

$$
\begin{aligned}
\int \psi_s(x, \omega)\psi_s(x', \omega) d\pi_s(\omega) &= \int \psi(x, \omega)\sqrt{C_s s(\omega)}\psi_s(x', \omega)\sqrt{C_s s(\omega)} \frac{1}{C_s s(\omega)} d\pi(\omega) \\
&= \int \psi(x, \omega)\psi(x', \omega) d\pi(\omega) = K(x, x').
\end{aligned}
$$

Now we show that $\psi_s, \pi_s$ achieves $\mathcal{F}_\infty(\lambda) = \mathcal{N}(\lambda)$. By recalling that $s(\omega) = \|(L + \lambda I)^{-1/2}\psi(\cdot, \omega)\|_{\rho_X}^{-2}$, we have

$$
\begin{aligned}
\mathcal{F}_\infty(\lambda) &= \sup_{\omega \in \Omega} \|(L + \lambda I)^{-1/2}\psi_s(\cdot, \omega)\|_{\rho_X}^2 = C_s \sup_{\omega \in \Omega} s(\omega)\|(L + \lambda I)^{-1/2}\psi(\cdot, \omega)\|_{\rho_X}^2 \\
&= C_s \sup_{\omega \in \Omega} \|(L + \lambda I)^{-1/2}\psi(\cdot, \omega)\|_{\rho_X}^{-2}\|(L + \lambda I)^{-1/2}\psi(\cdot, \omega)\|_{\rho_X}^2 = C_s.
\end{aligned}
$$

We recall that $C_s = \int \frac{1}{s(\omega)} d\pi$. Denoting with $\psi_\omega$ the function $\psi(\cdot, \omega)$ and considering that $\|Ax\|_{\rho_X} = \text{Tr}(A^2(x \otimes x))$ for any bounded symmetrix linear operator $A$ and vector $v$, and that the trace is linear,

$$
\begin{aligned}
\mathcal{F}_\infty(\lambda) = C_s &= \int \|(L + \lambda I)^{-1/2}\psi_\omega\|_{\rho_X}^2 d\pi(\omega) = \int \text{Tr}((L + \lambda I)^{-1}(\psi_\omega \otimes \psi_\omega)) d\pi(\omega) \\
&= \text{Tr}\left((L + \lambda I)^{-1} \int (\psi_\omega \otimes \psi_\omega) d\pi(\omega)\right) = \text{Tr}((L + \lambda I)^{-1}L) = \mathcal{N}(\lambda).
\end{aligned}
$$

where the fact that $L = \int \psi_\omega \otimes \psi_\omega d\pi(\omega)$ is due to Lemma 1. $\square$

**Proof of Theorem 3.** This is a version of Thm. 7, with simplified set of assumptions. Indeed the existence of $f_{\mathcal{H}}$ and the fact that $|y| \leq b$ a. s. satisfies Asm. 4 with $\sigma = B = 2b$. The fact that $X$ is a Polish space and that $\psi$ is bounded continuous, satisfy Asm. 3, and so the kernel

is bounded by $\kappa^2$. Asm. 4 and Asm. 6 are directly satisfied by Asm. 1. In particular we obtain
$$n_0 := (2/\|L\|)^{\frac{p+1}{p}} \vee \left(264\kappa^2 p \log(556\kappa^2\delta^{-1}\sqrt{p\kappa^2})\right)^{1+p},$$

$$c_0 := 9\left(3 + 4\kappa^2 + \frac{4\kappa^2}{\|L\|} + \frac{\kappa^2}{4}Q^{2r-1}F^{2-2r}\right), \quad c_1 := 64\left(\bar{B}\kappa + \bar{\sigma}Q + R\right)^2,$$

with $\bar{B} := 2b + 2\kappa R$ and $\bar{\sigma} := 2b + 2\kappa\sqrt{R}$. $\qquad\qquad\square$

# B   Concentration Inequalities

Here we recall some standard concentration inequalities that will be used in Sect. A.3. The following inequality is from Thm.3 of [31] and will be used in Lemma 10, together with other inequalities, to concentrate the empirical effective dimension to the true effective dimension.

**Proposition 1** (Bernstein's inequality for sum of random variables)*. Let $x_1, \ldots, x_n$ be a sequence of independent and identically distributed random variables on $\mathbb{R}$ with zero mean. If there exists an $T, S \in \mathbb{R}$ such that $x_i \leq T$ almost everywhere and $\mathbb{E}x_i^2 \leq S$, for $i \in \{1, \ldots, n\}$. For any $\delta > 0$ the following holds with probability at least $1 - \delta$:*

$$\frac{1}{n}\sum_{i=1}^{n} x_i \leq \frac{2T\log\frac{1}{\delta}}{3n} + \sqrt{\frac{2S\log\frac{1}{\delta}}{n}}.$$

*If there exists $T' \geq \max_i |x_i|$ almost everywhere, then the same bound, with $T'$ instead of $T$, holds for the for the absolute value of the left hand side, with probability at least $1 - 2\delta$.*

*Proof.* It is a restatement of Theorem 3 of [31]. $\qquad\qquad\square$

The following inequality is and adaptation of Thm. 3.3.4 in [32] and is a generalization of the previous one to random vectors. It is used primarily in Lemma 6, to control the sample error. Moreover it is used in Prop. 10, Lemma 10, to control the empirical effective dimension and to bound the term $\beta$ of Thm. 4, in the main theorem. Finally it is used to prove the inequality in Prop. 5.

**Proposition 2** (Bernstein's inequality for sum of random vectors)*. Let $z_1, \ldots, z_n$ be a sequence of independent identically distributed random vectors on a separable Hilbert space $\mathcal{H}$. Assume $\mu = \mathbb{E}z_i$ exists and let $\sigma, M \geq 0$ such that*

$$\mathbb{E}\|z_i - \mu\|_{\mathcal{H}}^p \leq \frac{1}{2}p!\sigma^2 M^{p-2}, \quad \forall p \geq 2,$$

*for any $i \in \{1, \ldots, n\}$. Then for any $\delta \in (0, 1]$:*

$$\left\|\frac{1}{n}\sum_{i=1}^{n} z_i - \mu\right\|_{\mathcal{H}} \leq \frac{2M\log\frac{2}{\delta}}{n} + \sqrt{\frac{2\sigma^2\log\frac{2}{\delta}}{n}}$$

*with probability at least $1 - \delta$.*

*Proof.* restatement of Theorem 3.3.4 of [32]. $\qquad\qquad\square$

The following inequality is essentially Thm. 7.3.1 in [33] (generalized to separable Hilbert spaces by the technique in Section 4 of [34]). It is a generalization of the Bernstein inequality to random operators. It is mainly used to prove the inequality in Prop. 6.

**Proposition 3** (Bernstein's inequality for sum of random operators)*. Let $\mathcal{H}$ be a separable Hilbert space and let $X_1, \ldots, X_n$ be a sequence of independent and identically distributed self-adjoint positive random operators on $\mathcal{H}$. Assume that there exists $\mathbb{E}X_i = 0$ and $\lambda_{\max}(X_i) \leq T$ almost surely for some $T > 0$, for any $i \in \{1, \ldots, n\}$. Let $S$ be a positive operator such that $\mathbb{E}(X_i)^2 \leq S$. Then for any $\delta \in (0, 1]$ the following holds*

$$\lambda_{\max}\left(\frac{1}{n}\sum_{i=1}^{n} X_i\right) \leq \frac{2T\beta}{3n} + \sqrt{\frac{2\|S\|\beta}{n}}$$

*with probability at least* $1 - \delta$. *Here* $\beta = \log \frac{2 \operatorname{Tr} S}{\|S\| \delta}$.

*If there exists* $L'$ *such that* $L' \geq \max_i \|X_i\|$ *almost everywhere, then the same bound holds with* $L'$ *instead of* $L$ *for the operator norm, with probability at least* $1 - 2\delta$.

*Proof.* The theorem is a restatement of Theorem 7.3.1 of [33] generalized to the separable Hilbert space case by means of the technique in Section 4 of [34]. □

## C   Operator Inequalities

Let $\mathcal{H}, \mathcal{K}$ be separable Hilbert spaces and $A, B : \mathcal{H} \to \mathcal{H}$ bounded linear operators.

The following inequality is needed to prove the interpolation inequality in Prop. 9, that is needed to perform a fine split of the computational error.

**Proposition 4** (Cordes Inequality [35]). *If* $A, B$ *are self-adjoint and positive, then*

$$\|A^s B^s\| \leq \|AB\|^s \quad when \ 0 \leq s \leq 1$$

## D   Auxiliary Results

The next proposition is used in Lemma 7 to control the sample error. It is based on the Bernstein inequality for random vectors, Prop. 2.

**Proposition 5.** *Let* $\mathcal{H}, \mathcal{K}$ *be two separable Hilbert spaces and* $(v_1, z_1), \ldots, (v_n, z_n) \in \mathcal{H} \times \mathcal{K}$, *with* $n \geq 1$, *be independent and identically distributed random pairs of vectors, such that there exists a constant* $\kappa > 0$ *for which* $\|v\|_{\mathcal{H}} \leq \kappa$ *and* $\|z\|_{\mathcal{H}} \leq \kappa$ *almost everywhere. Let* $Q = \mathbb{E}\, v \otimes v$, *let* $T = \mathbb{E}\, v \otimes z$ *and* $T_n = \frac{1}{n} \sum_{i=1}^{n} v_i \otimes z_i$. *For any* $0 < \lambda \leq \|Q\|$ *and any* $\tau \geq 0$, *the following holds*

$$\|(Q + \lambda I)^{-1/2}(T - T_n)\|_{HS} \leq \frac{4\sqrt{\tilde{\mathcal{F}}_\infty(\lambda)}\kappa \log \frac{2}{\tau}}{n} + \sqrt{\frac{4\kappa^2 \tilde{\mathcal{N}}(\lambda) \log \frac{2}{\tau}}{n}}$$

*with probability at least* $1 - \tau$, *where* ess sup *denotes the essential supremum and*

$$\tilde{\mathcal{F}}_\infty(\lambda) := \operatorname*{ess\,sup}_{v \in \mathcal{H}} \|(Q + \lambda I)^{-1/2}v\|^2, \quad \tilde{\mathcal{N}}(\lambda) := \operatorname{Tr}((Q + \lambda I)^{-1}Q).$$

*In particular, we recall that* $\tilde{\mathcal{N}}(\lambda) \leq \tilde{\mathcal{F}}_\infty(\lambda) \leq \frac{\kappa^2}{\lambda}$.

*Proof.* Define for any $i \in \{1, \ldots, n\}$ the random operator $\zeta_i = (Q + \lambda I)^{-1/2} v_i \otimes z_i$. Note that $\mathbb{E}\zeta_i = (Q + \lambda I)^{-1/2}T$. Since $\zeta_i$ is a vector in the Hilbert space of Hilbert-Schmidt operators on $\mathcal{H}$, we study the moments of $\|\zeta_i - \mathbb{E}\zeta_i\|_{HS}$ in order to apply Prop. 2. We recall that

$$\operatorname{ess\,sup}\|\zeta_i - \mathbb{E}\zeta_i\|_{HS} \leq \operatorname{ess\,sup}\|\zeta_i\|_{HS} + \mathbb{E}\|\zeta_i\|_{HS} \leq 2\operatorname{ess\,sup}\|\zeta_i\|_{HS}$$

$$= 2\operatorname{ess\,sup}\|(Q + \lambda I)^{-1/2}v_i \otimes z_i\|_{HS} \leq \operatorname{ess\,sup}\|(Q + \lambda I)^{-1/2}v_i\|_{\mathcal{H}}\|z_i\|_{\mathcal{K}}$$

$$\leq 2\operatorname{ess\,sup}\|(Q + \lambda I)^{-1/2}v_i\|_{\mathcal{H}}\ \operatorname{ess\,sup}\|z_i\|_{\mathcal{K}} = 2\tilde{F}_\infty(\lambda)^{1/2}\kappa.$$

For any $p \geq 2$ we have

$$\mathbb{E}\,\|\zeta_i - \mathbb{E}\zeta_i\|_{HS}^p \leq (\operatorname*{ess\,sup}_z\|\zeta_i - \mathbb{E}\zeta_i\|^{p-2})(\mathbb{E}\,\|\zeta_i - \mathbb{E}\zeta_i\|_{HS}^2)$$

$$\leq (2\tilde{F}_\infty(\lambda)^{1/2}\kappa)^{p-2}\mathbb{E}\|\zeta_i - \mathbb{E}\zeta_i\|_{HS}^2.$$

Now we study $\mathbb{E}\|\zeta_i - \mathbb{E}\zeta_i\|_{HS}^2$,

$$\mathbb{E}\|\zeta_i - \mathbb{E}\zeta_i\|_{HS}^2 = \operatorname{Tr}(\mathbb{E}\zeta_i \otimes \zeta_i - (\mathbb{E}\zeta_i)^2) \leq \operatorname{Tr}(\mathbb{E}\zeta_i \otimes \zeta_i)$$

$$= \mathbb{E}\ \|z_i\|^2 \operatorname{Tr}((Q + \lambda I)^{-1/2}(v_i \otimes v_i)(Q + \lambda I)^{-1/2})$$

$$\leq \operatorname{ess\,sup}\|z_i\|_{\mathcal{K}}^2\ \mathbb{E}\operatorname{Tr}((Q + \lambda I)^{-1/2}(v_i \otimes v_i)(Q + \lambda I)^{-1/2})$$

$$\leq \operatorname{ess\,sup}\|z_i\|_{\mathcal{K}}^2 \operatorname{Tr}((Q + \lambda I)^{-1/2}\mathbb{E}(v_i \otimes v_i)(Q + \lambda I)^{-1/2})$$

$$\leq \kappa^2 \operatorname{Tr}((Q + \lambda I)^{-1}Q) = \kappa^2 \tilde{\mathcal{N}}(\lambda),$$

for any $1 \le i \le n$. Therefore for any $p \ge 2$ we have

$$\mathbb{E}\,\|\zeta_i - \mathbb{E}\zeta_i\|_{HS}^p \le \frac{1}{2}p!\,\sqrt{2\kappa^2\mathcal{N}(\lambda)}^2 (2\tilde{F}_\infty(\lambda)^{1/2}\kappa)^{p-2}.$$

Finally we apply Prop. 2. $\qquad\qquad\qquad\qquad\qquad\qquad\qquad\qquad\qquad\qquad\qquad\square$

The following inequality, together with Prop. 8, is used in Prop. 10, Lemmas 10, 8. A similar technique can be found in [36].

**Proposition 6.** *Let $v_1, \dots, v_n$ with $n \ge 1$, be independent and identically distributed random vectors on a separable Hilbert spaces $\mathcal{H}$ such that $Q = \mathbb{E}\,v \otimes v$ is trace class, and for any $\lambda > 0$ there exists a constant $\mathcal{F}_\infty(\lambda) < \infty$ such that $\langle v, (Q + \lambda I)^{-1}v \rangle \le \mathcal{F}_\infty(\lambda)$ almost everywhere. Let $Q_n = \frac{1}{n}\sum_{i=1}^n v_i \otimes v_i$ and take $0 < \lambda \le \|Q\|$. Then for any $\delta \ge 0$, the following holds with probability at least $1 - 2\delta$*

$$\|(Q + \lambda I)^{-1/2}(Q - Q_n)(Q + \lambda I)^{-1/2}\| \le \frac{2\beta(1 + \mathcal{F}_\infty(\lambda))}{3n} + \sqrt{\frac{2\beta\mathcal{F}_\infty(\lambda)}{n}},$$

*where $\beta = \log \frac{4\,\mathrm{Tr}\,Q}{\lambda\delta}$. Moreover, with the same probability*

$$\lambda_{\max}\left((Q + \lambda I)^{-1/2}(Q - Q_n)(Q + \lambda I)^{-1/2}\right) \le \frac{2\beta}{3n} + \sqrt{\frac{2\beta\mathcal{F}_\infty(\lambda)}{n}}.$$

*Proof.* Let $Q_\lambda = Q + \lambda I$. Here we apply Prop. 3 on the random variables $Z_i = M - Q_\lambda^{-1/2}v_i \otimes Q_\lambda^{-1/2}v_i$ with $M = Q_\lambda^{-1/2}QQ_\lambda^{-1/2}$ for $1 \le i \le n$. Note that the expectation of $Z_i$ is 0. The random vectors are bounded by

$$\|Q_\lambda^{-1/2}QQ_\lambda^{-1/2} - Q_\lambda^{-1/2}v_i \otimes Q_\lambda^{-1/2}v_i\| \le \langle v_i, Q_\lambda^{-1}v_i \rangle + \|Q_\lambda^{-1/2}QQ_\lambda^{-1/2}\| \le \mathcal{F}_\infty(\lambda) + 1,$$

almost everywhere, for any $1 \le i \le n$. The second order moment is

$$\begin{aligned}
\mathbb{E}(Z_i)^2 = \mathbb{E}\ \langle v_i, Q_\lambda^{-1}v_i \rangle\ Q_\lambda^{-1/2}v_i \otimes Q_\lambda^{-1/2}v_i \quad &- \quad Q_\lambda^{-2}Q^2 \\
&\le \mathcal{F}_\infty(\lambda)\mathbb{E}Q_\lambda^{-1/2}v_i \otimes Q_\lambda^{-1/2}v_i = \mathcal{F}_\infty(\lambda)Q = S,
\end{aligned}$$

for $1 \le i \le n$. Now we can apply Prop. 3. Now some considerations on $\beta$. It is $\beta = \log \frac{2\,\mathrm{Tr}\,S}{\|S\|\delta} = \frac{2\,\mathrm{Tr}\,Q_\lambda^{-1}Q}{\|Q_\lambda^{-1}Q\|\delta}$, now $\mathrm{Tr}\,Q_\lambda^{-1}Q \le \frac{1}{\lambda}\,\mathrm{Tr}\,Q$. We need a lower bound for $\|Q_\lambda^{-1}Q\| = \frac{\sigma_1}{\sigma_1 + \lambda}$ where $\sigma_1 = \|Q\|$ is the biggest eigenvalue of $Q$, now $\lambda \le \sigma_1$ thus $\frac{\sigma_1}{\sigma_1+\lambda} \ge 1/2$ and so $\beta \le \log \frac{2\,\mathrm{Tr}\,Q_\lambda^{-1}Q}{\|Q_\lambda^{-1}Q\|\delta} \le \log \frac{4\,\mathrm{Tr}\,Q}{\lambda\delta}$.

For the second bound of this proposition we use the second bound of Prop. 3, the analysis remains the same except for uniform bound on $Z_1$, that now is

$$\sup_{f \in \mathcal{H}} \langle f, Z_1 f \rangle = \sup_{f \in \mathcal{H}} \langle f, Q_\lambda^{-1}Qf \rangle - \left\langle f, Q_\lambda^{-1/2}v_i \right\rangle^2 \le \sup_{f \in \mathcal{H}} \langle f, Q_\lambda^{-1}Qf \rangle \le 1.$$

$\qquad\qquad\qquad\qquad\qquad\qquad\qquad\qquad\qquad\qquad\qquad\qquad\qquad\qquad\qquad\qquad\square$

In the following remark, we start from the result of the previous proposition, expressing the conditions on $n$ and $\lambda$ with respect to a given value for the bound.

**Remark 10.** *With the same notation of Prop. 6, assume that $\|v\| \le \kappa$ almost everywhere[3], then we have that*

1. *for any $t \in (0, 1]$, when $n \ge \frac{2}{t^2}\left(\frac{2t}{3} + \mathcal{F}_\infty(\lambda)\right)\log\frac{4\kappa^2}{\lambda\delta}$ and $\lambda \le \|Q\|$, we have*

$$\lambda_{\max}\left((Q + \lambda I)^{-1/2}(Q - Q_n)(Q + \lambda I)^{-1/2}\right) \le t,$$

   *with probability at least $1 - \delta$.*

2. *The equation above holds with the same probability with $t = 1/2$, when $\frac{9\kappa^2}{n} \log \frac{n}{2\delta} \leq \lambda \leq \|Q\|$ and $n \geq 405\kappa^2 \vee 67\kappa^2 \log \frac{\kappa^2}{2\delta}$.*

3. *The equation above holds with the same probability with $t = 1/3$, when $\frac{19\kappa^2}{n} \log \frac{n}{4\delta} \leq \lambda \leq \|Q\|$ and $n \geq 405\kappa^2 \vee 67\kappa^2 \log \frac{\kappa^2}{2\delta}$.*

The next proposition, together with Prop. 10 are a restatement of Prop. 1 of [16]. In particular the next proposition performs the analytic decomposition of the difference between the empirical and the true effective dimension, while Prop. 10, bounds the decomposition in probability.

**Proposition 7** (Geometry of Empirical Effective Dimension). *Let $\mathcal{L}$ be an Hilbert space. Let $L, L_M : \mathcal{L} \to \mathcal{L}$ be two bounded positive linear operators, that are trace class. Given $\lambda > 0$, let*

$$\mathcal{N}(\lambda) = \mathrm{Tr}(LL_\lambda^{-1}) \quad \text{and} \quad \tilde{\mathcal{N}}(\lambda) = \mathrm{Tr}(L_M L_{M,\lambda}^{-1}),$$

*with $L_\lambda = L + \lambda I$. Then*

$$|\tilde{\mathcal{N}}(\lambda) - \mathcal{N}(\lambda)| \leq \lambda e(\lambda) + \frac{d(\lambda)^2}{1 - c(\lambda)}$$

*where $c(\lambda) = \lambda_{\max}(\tilde{B})$, $d(\lambda) = \|\tilde{B}\|_{\mathrm{HS}}$ with $\tilde{B} = L_\lambda^{-1/2}(L - L_M)L_\lambda^{-1/2}$ and $e(\lambda) = \mathrm{Tr}(L_\lambda^{-1/2}\tilde{B}L_\lambda^{-1/2})$.*

*Proof.* First of all note that $\lambda_{\max}(\tilde{B})$, the biggest eigenvalue of $\lambda_{\max}$, is smaller than 1 since $\tilde{B}$ is the difference of two positive operators and the biggest eigenvalue of the minuend operator is $\|L_\lambda^{-1/2}LL_\lambda^{-1/2}\| = \frac{\lambda_{\max}(L)}{\lambda_{\max}(L)+\lambda} < 1$. Then we can use the fact that $A(A+\lambda I)^{-1} = I - \lambda(A+\lambda I)^{-1}$ for any bounded linear operator $A$ and that $L_{M,\lambda}^{-1} = L_\lambda^{-1/2}(I - \tilde{B})^{-1}L_\lambda^{-1/2}$ (see the proof of Prop. 8), since $\lambda_{\max}(\tilde{B}) < 1$, to obtain

$$|\tilde{\mathcal{N}}(\lambda) - \mathcal{N}(\lambda)| = |\mathrm{Tr}(L_{M,\lambda}^{-1}L_M - LL_\lambda^{-1})| = \lambda|\mathrm{Tr}(L_{M,\lambda}^{-1} - L_\lambda^{-1})| = |\lambda\,\mathrm{Tr}(L_{M,\lambda}^{-1}(L_M - L)L_\lambda^{-1})|$$

$$= |\lambda\,\mathrm{Tr}(L_\lambda^{-1/2}(I - \tilde{B})^{-1}L_\lambda^{-1/2}(L_M - L)L_\lambda^{-1/2}L_\lambda^{-1/2})|$$

$$= |\lambda\,\mathrm{Tr}(L_\lambda^{-1/2}(I - \tilde{B})^{-1}\tilde{B}L_\lambda^{-1/2})|.$$

Considering that for any bounded symmetric linear operator $X$ the following identity holds

$$(I - X)^{-1}X = X + X(I - X)^{-1}X,$$

when $\lambda_{\max}(X) < 1$, we have

$$\lambda|\mathrm{Tr}(L_\lambda^{-1/2}(I - \tilde{B})^{-1}\tilde{B}L_\lambda^{-1/2})| \leq \underbrace{\lambda|\mathrm{Tr}(L_\lambda^{-1/2}\tilde{B}L_\lambda^{-1/2})|}_{A} + \underbrace{\lambda|\mathrm{Tr}(L_\lambda^{-1/2}\tilde{B}(I - \tilde{B})^{-1}\tilde{B}L_\lambda^{-1/2})|}_{B}.$$

The term $A$ is just equal to $\lambda e(\lambda)$. Now, by definition of Hilbert-Schmidt norm, the term $B$ can be written as $B = \|\lambda^{1/2}L_\lambda^{-1/2}\tilde{B}(I - \tilde{B})^{-1/2}\|_{\mathrm{HS}}^2$, thus we have

$$B = \|\lambda^{1/2}L_\lambda^{-1/2}\tilde{B}(I - \tilde{B})^{-1/2}\|_{\mathrm{HS}}^2 \leq \|\lambda^{1/2}L_\lambda^{-1/2}\|^2\|\tilde{B}\|_{\mathrm{HS}}^2\|(I - \tilde{B})^{-1/2}\|^2 \leq (1-c(\lambda))^{-1}d(\lambda)^2,$$

since $\|(I - \tilde{B})^{-1/2}\|^2 = (1 - \lambda_{\max}(\tilde{B}))^{-1}$ because the spectral function $(1 - \sigma)^{-1}$ is increasing and positive on $[-\infty, 1)$. $\qquad\square$

The next result is essentially Prop. 7 of [16], while a similar technique can be found in [36]. It is used, mainly together with Prop. 6, to give multiplicative bounds to empirical operators. It is used in the analytic decomposition of the excess risk, in Prop. 7, Thm. 4.

**Proposition 8.** *Let $\mathcal{H}$ be a separable Hilbert space, let $A, B$ two bounded self-adjoint positive linear operators on $\mathcal{H}$ and $\lambda > 0$. Then*

$$\|(A + \lambda I)^{-1/2}B^{1/2}\| \leq \|(A + \lambda I)^{-1/2}(B + \lambda I)^{1/2}\| \leq (1 - \beta)^{-1/2}$$

*with*

$$\beta = \lambda_{\max}\left[(B + \lambda I)^{-1/2}(B - A)(B + \lambda I)^{-1/2}\right].$$

*Note that $\beta \leq \frac{\lambda_{\max}(B)}{\lambda_{\max}(B)+\lambda} < 1$ by definition.*

*Proof.* Let $B_\lambda = B + \lambda I$. First of all note that $\beta < 1$ for any $\lambda > 0$. Indeed, by exploiting the variational formulation of the biggest eigenvalue, we have

$$\beta = \lambda_{\max}(B_\lambda^{-1/2}(B-A)B_\lambda^{-1/2}) = \sup_{f \in \mathcal{H}, \|f\|_\mathcal{H} \leq 1} \left\langle f, B_\lambda^{-1/2}(B-A)B_\lambda^{-1/2}f \right\rangle$$

$$= \sup_{f \in \mathcal{H}, \|f\|_\mathcal{H} \leq 1} \left\langle f, B_\lambda^{-1/2}BB_\lambda^{-1/2}f \right\rangle - \left\langle f, B_\lambda^{-1/2}AB_\lambda^{-1/2}f \right\rangle$$

$$\leq \sup_{f \in \mathcal{H}, \|f\|_\mathcal{H} \leq 1} \left\langle f, B_\lambda^{-1/2}BB_\lambda^{-1/2}f \right\rangle = \lambda_{\max}(B_\lambda^{-1/2}BB_\lambda^{-1/2})$$

$$\leq \frac{\lambda_{\max}(B)}{\lambda_{\max}(B) + \lambda} < 1,$$

since $A$ is a positive operator and thus $\left\langle f, B_\lambda^{-1/2}AB_\lambda^{-1/2}f \right\rangle \geq 0$ for any $f \in \mathcal{H}$. Now note that

$$(A + \lambda I)^{-1} = [(B + \lambda I) - (B - A)]^{-1}$$

$$= \left[ B_\lambda^{1/2} \left( I - B_\lambda^{-1/2}(B-A)B_\lambda^{-1/2} \right) B_\lambda^{1/2} \right]^{-1}$$

$$= B_\lambda^{-1/2} \left[ I - B_\lambda^{-1/2}(B-A)B_\lambda^{-1/2} \right]^{-1} B_\lambda^{-1/2}.$$

Now let $X = (I - B_\lambda^{-1/2}(B-A)B_\lambda^{-1/2})^{-1}$. We have that,

$$\|(A + \lambda I)^{-1/2}B_\lambda^{1/2}\| = \|B_\lambda^{1/2}(A + \lambda I)^{-1}B_\lambda^{1/2}\|^{1/2} = \|X\|^{1/2}$$

because $\|Z\| = \|Z^*Z\|^{1/2}$ for any bounded operator $Z$. Note that

$$\|(A + \lambda I)^{-1/2}B^{1/2}\| = \|(A + \lambda I)^{-1/2}B_\lambda^{1/2}B_\lambda^{-1/2}B^{1/2}\| \leq \|(A + \lambda I)^{-1/2}B_\lambda^{1/2}\|\|B_\lambda^{-1/2}B^{1/2}\|$$

$$\leq \|X\|^{1/2}\|B_\lambda^{-1/2}B^{1/2}\| \leq \|X\|^{1/2}.$$

Finally let $Y = B_\lambda^{-1/2}(B-A)B_\lambda^{-1/2}$, we have seen that $\beta = \lambda_{\max}(Y) < 1$, then

$$\|X\| = \|(I - Y)^{-1}\| = (1 - \lambda_{\max}(Y))^{-1},$$

since $X = w(Y)$ with $w(\sigma) = (1 - \sigma)^{-1}$ for $-\infty \leq \sigma < 1$, and $w$ is positive and monotonically increasing on the domain. $\qquad\square$

The following proposition is used to give a fine analytical decomposition of the excess risk in Prop. 7, Thm. 4. A similar interpolation inequality for finite dimensional matrices, can be found in [37]. Here we prove it for bounded linear operators on separable Hilbert spaces.

**Proposition 9.** *Let $\mathcal{H}, \mathcal{K}$ be two separable Hilbert spaces and $X, A$ be bounded linear operators, with $X : \mathcal{H} \to \mathcal{K}$ and $A : \mathcal{H} \to \mathcal{H}$ be positive semidefinite.*

$$\|XA^\sigma\| \leq \|X\|^{1-\sigma}\|XA\|^\sigma, \qquad \forall \sigma \in [0,1]. \tag{44}$$

*Proof.*

$$\|XA^\sigma\| = \|A^\sigma(X^*X)^{\frac{1}{\sigma}\sigma}A^\sigma\|^{\frac{1}{2}} \leq \|A(X^*X)^{\frac{1}{\sigma}}A\|^{\frac{\sigma}{2}} \tag{45}$$

where the last inequality is due to Cordes (see Proposition 4). Then we have that $(X^*X)^{\frac{1}{\sigma}} \preccurlyeq \|X\|^{\frac{2(1-\sigma)}{\sigma}}X^*X$ (where $\preccurlyeq$ is the Löwner partial ordering on positive operators) and so

$$A(X^*X)^{\frac{1}{\sigma}}A \preccurlyeq \|X\|^{\frac{2(1-\sigma)}{\sigma}}AX^*XA$$

that implies

$$\|A(X^*X)^{\frac{1}{\sigma}}A\|^{\frac{\sigma}{2}} \leq \|X\|^{1-\sigma}\|AX^*XA\|^{\frac{\sigma}{2}} = \|X\|^{1-\sigma}\|XA\|^\sigma \tag{46}$$

$\qquad\square$

In the next proposition, we bound $\mathcal{N}_M(\lambda)$ in terms of the effective dimension $\mathcal{N}(\lambda)$ that is the one associated to the kernel $K$. The proof of Prop. 10 analogous to the one of Prop. 1 of [16], with simpler proof and slightly improved constants.

**Proposition 10** (Bounds on the Effective Dimension). *Let $\tilde{\mathcal{N}}(\lambda) = \operatorname{Tr} L_M L_{M,\lambda}^{-1}$. Under Assumption 3, for any $\delta > 0$, $\lambda \leq \|L\|$ and $m \geq (4 + 18\mathcal{F}_\infty(\lambda)) \log \frac{12\kappa^2}{\lambda\delta}$, then the following holds with probability at least $1 - \delta$,*

$$|\tilde{\mathcal{N}}(\lambda) - \mathcal{N}(\lambda)| \leq c(\lambda, m, \delta)\mathcal{N}(\lambda) \leq 1.55\mathcal{N}(\lambda),$$

*with $c(\lambda, m, \delta) = \frac{q}{\mathcal{N}(\lambda)} + \sqrt{\frac{3q}{2\mathcal{N}(\lambda)}} + \frac{3}{2}\left(\frac{3q}{\sqrt{\mathcal{N}(\lambda)}} + \sqrt{3q}\right)^2$ and $q = \frac{4\mathcal{F}_\infty(\lambda)\log\frac{6}{\delta}}{3m}$.*

*Proof.* First of all we recall that $\tilde{\mathcal{N}}(\lambda) = \operatorname{Tr} L_M L_{M,\lambda}^{-1}$ and that $\mathcal{N}(\lambda) = \operatorname{Tr} L L_\lambda^{-1}$. Let $\tau = \delta/3$. By Prop. 7 we know that

$$|\tilde{\mathcal{N}}(\lambda) - \mathcal{N}(\lambda)| \leq \left(\frac{d(\lambda)^2}{(1 - c(\lambda))\mathcal{N}(\lambda)} + \frac{\lambda e(\lambda)}{\mathcal{N}(\lambda)}\right)\mathcal{N}(\lambda)$$

where $c(\lambda) = \lambda_{\max}(\tilde{B})$, $d(\lambda) = \|\tilde{B}\|_{\mathrm{HS}}$ with $\tilde{B} = L_\lambda^{-1/2}(L - L_M)L_\lambda^{-1/2}$ and $e(\lambda) = |\operatorname{Tr}(L_\lambda^{-1/2}\tilde{B}L_\lambda^{-1/2})|$. Thus, now we control $c(\lambda)$, $d(\lambda)$ and $e(\lambda)$ in probability. Choosing $m$ such that $m \geq (4 + 18\mathcal{F}_\infty(\lambda))\log\frac{4\kappa^2}{\lambda\tau}$, Prop. 6 guarantees that $c(\lambda) = \lambda_{\max}(\tilde{B}) \leq 1/3$ with probability at least $1 - \tau$.

To find an upper bound for $\lambda e(\lambda)$ we define the i.i.d. random variables $\eta_i = \langle \psi_{\omega_i}, \lambda L_\lambda^{-2}\psi_{\omega_i}\rangle_{\rho_X} \in \mathbb{R}$ with $i \in \{1, \ldots, m\}$. By linearity of the trace and the expectation, we have $M = \mathbb{E}\eta_1 = \mathbb{E}\langle \psi_{\omega_1}, \lambda L_\lambda^{-2}\psi_{\omega_1}\rangle_{\rho_X} = \mathbb{E}\operatorname{Tr}(\lambda L_\lambda^{-2}\psi_{\omega_1} \otimes \psi_{\omega_1}) = \lambda\operatorname{Tr}(L_\lambda^{-2}L)$. Therefore,

$$\lambda e(\lambda, m) = \left|\lambda\operatorname{Tr}(L_\lambda^{-1/2}\tilde{B}L_\lambda^{-1/2})\right| = \left|\lambda\operatorname{Tr}\left(L_\lambda^{-1}LL_\lambda^{-1} - \frac{1}{m}\sum_{i=1}^m (L_\lambda^{-1}\psi_{\omega_i}) \otimes (L_\lambda^{-1}\psi_{\omega_i})\right)\right|$$

$$= \left|\operatorname{Tr}\left(\lambda L_\lambda^{-1}LL_\lambda^{-1}\right) - \frac{1}{m}\sum_{i=1}^m \langle \psi_{\omega_i}, \lambda L_\lambda^{-2}\psi_{\omega_i}\rangle_{\rho_X}\right| = \left|M - \frac{1}{m}\sum_{i=1}^m \eta_i\right|.$$

By noting that $M$ is upper bounded by $M = \operatorname{Tr}(\lambda L_\lambda^{-2}L) = \operatorname{Tr}((I - L_\lambda^{-1}L)L_\lambda^{-1}L) \leq \mathcal{N}(\lambda)$, we can apply the Bernstein inequality (Prop. 1) where $T$ and $S$ are

$$\sup_{\omega \in \Omega}|M - \eta_1| \leq \lambda\|L_\lambda^{-1/2}\|^2\|L_\lambda^{-1/2}\psi_{\omega_1}\|^2 + M \leq \mathcal{F}_\infty(\lambda) + \mathcal{N}(\lambda) \leq 2\mathcal{F}_\infty(\lambda) = T,$$

$$\mathbb{E}(\eta_1 - M)^2 = \mathbb{E}\eta_1^2 - M^2 \leq \mathbb{E}\eta_1^2 \leq (\sup_{\omega\in\Omega}|\eta_1|)(\mathbb{E}\eta_1) \leq \mathcal{F}_\infty(\lambda)\mathcal{N}(\lambda) = S.$$

Thus, we have

$$\lambda|\operatorname{Tr} L_\lambda^{-1/2}\tilde{B}L_\lambda^{-1/2}| \leq \frac{4\mathcal{F}_\infty(\lambda)\log\frac{2}{\tau}}{3m} + \sqrt{\frac{2\mathcal{F}_\infty(\lambda)\mathcal{N}(\lambda)\log\frac{2}{\tau}}{m}},$$

with probability at least $1 - \tau$.

To find a bound for $d(\lambda)$ consider that $\tilde{B} = W - \frac{1}{m}\sum_{i=1}^m \zeta_i$ where $\zeta_i$ are i.i.d. random operators defined as $\zeta_i = L_\lambda^{-1/2}(\psi_{\omega_i} \otimes \psi_{\omega_i})L_\lambda^{-1/2} \in \mathcal{L}$ for all $i \in \{1, \ldots, m\}$, and $W = \mathbb{E}\zeta_1 = L_\lambda^{-1}L \in \mathcal{L}$. Then, by noting that $\|W\|_{HS} \leq \mathbb{E}\operatorname{Tr}(\zeta_1) = \mathcal{N}(\lambda)$, we can apply the Bernstein's inequality for random vectors on a Hilbert space (Prop. 2) where $T$ and $S$ are:

$$\|W - \zeta_1\|_{HS} \leq \|L_\lambda^{-1/2}\psi_{\omega_1}\|_{\rho_X}^2 + \|W\|_{HS} \leq \mathcal{F}_\infty(\lambda) + \|W\|_{HS} \leq 2\mathcal{F}_\infty(\lambda) = T,$$

$$\mathbb{E}\|\zeta_1 - W\|^2 = \mathbb{E}\operatorname{Tr}(\zeta_1^2 - W^2) \leq \mathbb{E}\operatorname{Tr}(\zeta_1^2) \leq (\sup_{\omega\in\Omega}\|\zeta_1\|)(\mathbb{E}\operatorname{Tr}(\zeta_1)) = \mathcal{F}_\infty(\lambda)\mathcal{N}(\lambda) = S,$$

obtaining

$$d(\lambda) = \|\tilde{B}\|_{HS} \leq \frac{4\mathcal{F}_\infty(\lambda)\log\frac{2}{\tau}}{m} + \sqrt{\frac{4\mathcal{F}_\infty(\lambda)\mathcal{N}(\lambda)\log\frac{2}{\tau}}{m}},$$

with probability at least $1 - \tau$. Then, by taking a union bound for the three events we have

$$|\tilde{\mathcal{N}}(\lambda) - \mathcal{N}(\lambda)| \leq \left( \frac{q}{\mathcal{N}(\lambda)} + \sqrt{\frac{3q}{2\mathcal{N}(\lambda)}} + \frac{3}{2} \left( \frac{3q}{\sqrt{\mathcal{N}(\lambda)}} + \sqrt{3q} \right)^2 \right) \mathcal{N}(\lambda),$$

with probability at least $1 - \delta$, where $q = \frac{4\mathcal{F}_\infty(\lambda)\log\frac{6}{\delta}}{3m}$. Finally, if $m \geq (4 + 18\mathcal{F}_\infty(\lambda))\log\frac{12\kappa^2}{\lambda\delta}$, then we have $q \leq 2/27$. Noting that $\mathcal{N}(\lambda) \geq \|LL_\lambda^{-1}\| = \frac{\|L\|}{\|L\|+\lambda} \geq 1/2$, we have that

$$|\tilde{\mathcal{N}}(\lambda) - \mathcal{N}(\lambda)| \leq 1.55\mathcal{N}(\lambda).$$

$\square$

# E    Examples of Random feature maps

A lot of works have been devoted to develop random feature maps in the the setting introduced above, or slight variations (see for example [13, 38, 20, 25, 39, 40, 41, 42, 43, 44] and references therein). In the rest of the section, we give several examples.

**Random Features for Translation Invariant Kernels and extensions [13, 38, 41, 44]**    This approximation method is defined in [13] for the translation invariant kernels when $X = \mathbb{R}^d$. A kernel $k : X \times X \to \mathbb{R}$ is *translation invariant*, when there exists a function $v : X \to \mathbb{R}$ such that $k(x,z) = v(x - z)$ for all $x, z \in X$. Now, let $\hat{v} : \Omega \to \mathbb{R}$ be the Fourier transform of $v$, with $\Omega = X \times [0, 2\pi]$. As shown in [13], by using the Fourier transform, we can express $k$ as

$$k(x, z) = v(x - z) = \int_\Omega \psi(\omega, x)\psi(\omega, z)\hat{v}(\omega)d\pi, \quad \forall x, z \in X,$$

with $\pi$ proportional to $\hat{v}$, $\psi(\omega, x) = \cos(w^\top x + b)$ and $\omega := (w, b) \in \Omega,\ x \in X$. Note that $X, \Omega, \pi, \psi$ satisfy Assumption 3.
In [38], they further randomize the construction above, by using results from locally sensitive hashing, to obtain a feature map which is a binary code. It can be shown that their methods satisfy Assumption 3 for an appropriate choice of $\Omega$ and the probability distribution $\pi$.
[41, 44] consider the setting of [13], but [41] selects $\omega_1, \ldots, \omega_m$ by means of the fast Welsh-Hadamard transform in order to improve the computational complexity for the algorithm computing $K_M$, while [44] selects them by using low-discrepancy sequences for quasi-random sampling to improve the statistical accuracy of $K_M$ with respect to $K$.

**Random Features Maps for the Gaussian Kernel, which are functions in $\mathcal{H}$.**    This set of random features is related to the deterministic polynomial features in [45]. Let $X = [0, 1]^d$ and $\Omega = \mathbb{N}^d$. Let $\sigma > 0$. We have

$$\psi(\omega, x) = C^{1/2}e^{-\frac{\sigma^2}{2}\|x\|^2}\prod_{j=1}^{d} x_j^{\omega_j}, \qquad \pi(\omega) = \frac{\sigma^{-2\sum_j \omega_j}}{C\,\omega_1! \ldots \omega_d!},$$

where $C$ is the normalization factor $C = e^{\sigma^2 d}$. Indeed by Taylor expansion of the exponential function and of the power of a multinomial, we have

$$e^{-\frac{\sigma^2}{2}\|x-x'\|^2} = e^{-\frac{\sigma^2}{2}(\|x\|^2+\|x'\|^2)}e^{\sigma^2 x^\top x} = e^{-\frac{\sigma^2}{2}(\|x\|^2+\|x'\|^2)}\sum_{t\geq 0}\frac{\sigma^{2t}}{t!}(x^\top x')^t$$

$$= e^{-\frac{\sigma^2}{2}(\|x\|^2+\|x'\|^2)}\sum_{t\geq 0}\frac{\sigma^{2t}}{t!}(\sum_{j=1}^{d} x_j x'_j)^t$$

$$= e^{-\frac{\sigma^2}{2}(\|x\|^2+\|x'\|^2)}\sum_{t\geq 0}\frac{\sigma^{2t}}{t!}\sum_{\omega_1+\cdots+\omega_d=t}\frac{t!}{\omega_1!\ldots\omega_d!}\prod_{j=1}^{d}(x_j x'_j)^{\omega_j}$$

$$= \sum_{t\geq 0}\frac{\sigma^{2t}}{Ct!}\sum_{\omega_1+\cdots+\omega_d=t}\frac{t!}{\omega_1!\ldots\omega_d!}\left(C^{1/2}e^{-\frac{\sigma^2}{2}\|x\|^2}\prod_{j=1}^{d}x_j^{\omega_j}\right)\left(C^{1/2}e^{-\frac{\sigma^2}{2}\|x'\|^2}\prod_{j=1}^{d}x_j'^{\omega_j}\right)$$

$$= \sum_{\omega\in\mathbb{N}^d}\psi(x,\omega)\psi(x',\omega)\pi(\omega).$$

Note that it is possible to sample from $\pi$ in the following way. By the steps above it is clear that a sample from $\pi$ can be obtained by first sampling $t$ from a Poisson distribution with parameter $\sigma^2 d$ and then sampling $\omega_1,\ldots,\omega_d$ from a multinomial distribution with probability $p_1 = \cdots = p_d = 1/d$ and number of trials $t$.

Finally note that $\psi(\omega,\cdot)$ is in $\mathcal{H}$, the RKHS induced by the Gaussian kernel (to prove it apply Prop. 3.6 of [46] with $b_\nu = \delta_{\nu=(\omega_1,\ldots,\omega_d)}$ and note that $e_\nu$ is exactly a multiple of $\psi(\omega,\cdot)$). Additionally note that $\sup_{\omega,x}|\psi(\omega,x)| < \infty$ and moreover $\|\psi(\omega,\cdot)\|_\mathcal{H} < \infty$ for any $\omega \in \mathbb{N}^d$. However $\lim_{\|\omega\|\to\infty}\|\psi(\omega,\cdot)\|_\mathcal{H} = \infty$.

**Random Features Maps for Dot Product Kernels [39, 40, 43]**  This approximation method is defined for the dot product kernels when $X$ is the ball of $\mathbb{R}^d$ of radius $R$, with $R > 0$. The considered kernels are of the form $k(x,z) = v(x^\top z)$ for a $v : \mathbb{R} \to \mathbb{R}$ such that

$$v(t) = \sum_{p=0}^{\infty} c_p t^p, \quad \text{with} \quad c_p \geq 0 \; \forall p \geq 0.$$

[39], start from the consideration that when $w \in \{-1,1\}^d$ is a vector of $d$ independent random variables with probability at least $1/2$, then $\mathbb{E}\, ww^\top = I$. Thus

$$\mathbb{E}_w\,(x^\top w)(z^\top w) = \mathbb{E}\, x^\top(ww^\top)z = x^\top\mathbb{E}(ww^\top)z = x^\top z,$$

and $(x^\top z)^p$ can be approximated by $g(W_p,x)^\top g(W_p,z)$ with $g(W_p,x) = \prod_{i=1}^{p} x^\top w_i$ and $W_p = (w_1,\ldots,w_p) \in \{-1,1\}^{p\times d}$ a matrix of independent random variables with probability at least $1/2$. Indeed it holds,

$$\mathbb{E}_{W_p}\, g(W_p,x)g(W_p,z) = \mathbb{E}_{W_p}\prod_{i=1}^{p}(x^\top w_i)\prod_{i=1}^{p}(z^\top w_i) = \mathbb{E}_{W_p}\prod_{i=1}^{p}(x^\top w_i)(z^\top w_i)$$

$$= \prod_{i=1}^{p}\mathbb{E}_{w_i}\,(x^\top w_i)(z^\top w_i) = \prod_{i=1}^{p} x^\top z = (x^\top z)^p.$$

Therefore the idea is to define the following sample space $\Omega = \mathbb{N}_0 \times (\bigcup_{p=1}^{\infty} T^p)$ with $T^p = \{-1,1\}^{p\times d}$, the probability $\pi$ on $\Omega$ as $\pi(p,W_q) = \pi_\mathbb{N}(p)\pi(W_q|p)$ for any $p,q \in \mathbb{N}_0$, $W_q \in T^q$ with $\pi_\mathbb{N}(p) = \frac{\tau^{-p-1}}{\tau-1}$ for a $\tau > 1$ and $\pi(W_q|p) = 2^{-pd}\delta_{pq}$ and the function

$$\psi(\omega,x) = \sqrt{c_p \tau^{p+1}(\tau-1)}g(W_p,x), \quad \forall\omega = (p,W_p) \in \Omega$$

where $w_i$ is the $i$-th row of $W_p$ with $1 \leq i \leq p$. Now note that Eq. (6) is satisfied, indeed for any $x, z \in X$

$$\int_\Omega \psi(\omega, x)\psi(\omega, z)d\pi = \int_{\mathbb{N}_0 \times (\bigcup_{p=1}^\infty T^p)} c_p g(W_p, x)g(W_p, z)dp d\pi(W_q|p)$$

$$= \sum_{p=0}^\infty c_p \, \mathbb{E}_{W_p} g(W_p, x)g(W_p, z)$$

$$= \sum_{p=0}^\infty c_p \, (x^\top z)^p = v(x^\top z) = K(x, z).$$

Now note that Assumption 3 is satisfied when $v(\tau R^2 d) < \infty$. Indeed, considering that $(x^\top w)^2 \leq \|x\|^2 \|w\|^2 \leq R^2 d$ for any $x \in X$ and $w \in \{-1, 1\}^d$, we have

$$\sup_{\omega \in \Omega, x \in X} |\psi(\omega, x)|^2 = \sup_{p \in \mathbb{N}_0, W_p \in T^p, x \in X} c_p \tau^{p+1}(\tau - 1)g_p(W_p, x)g_p(W_p, z)$$

$$= \sup_{p \in \mathbb{N}_0} c_p \tau^{p+1}(\tau - 1) \sup_{W_p \in T^p, x \in X} \prod_{i=1}^p (x^\top w_i)^2$$

$$\leq \sup_{p \in \mathbb{N}_0} c_p \tau^{p+1}(\tau - 1)(R^2 d)^p \leq \frac{\tau}{\tau - 1} \sum_{p=0}^\infty c_p \tau^p (R^2 d)^p$$

$$= \frac{\tau}{\tau - 1} v(\tau R^2 d).$$

[40, 43] approximate the construction above by using randomized tensor sketching and Johnson-Lindenstrauss random projections. It can be shown that even their methods satisfy Assumption 3 for an appropriate choice of $\Omega$ and the probability distribution $\pi$.

**Random Laplace Feature Maps for Semigroup invariant Kernels [42]**  The considered input space is $X = [0, \infty)^d$ and the considered kernels are of the form

$$K(x, z) = v(x + z), \quad \forall x, z \in X,$$

and $v : X \to \mathbb{R}$ is a function that is positive semidefinite. By Berg's theorem, it is equivalent to the fact that $\breve{v}$, the Laplace transform of $v$ is such that $\breve{v}(\omega) \geq 0$, for all $\omega \in X$ and that $\int_X \breve{v}(\omega)d\omega = V < \infty$. It means that we can express $K$ by Eq. (6), where $\Omega = X$, the feature map is $\psi(\omega, x) = \sqrt{V} e^{-\omega^\top x}$, for all $\omega \in \Omega, x \in X$ and the probability density is $\pi(\omega) = \frac{\breve{v}(\omega)}{V}$. Note that Assumption 3 is satisfied.

**Homogeneous additive Kernels [47]**  In this work they focus on $X = [0, 1]^d$ and on additive homogeneous kernels, that are of the form

$$K(x, z) = \sum_{i=1}^d k(x_i, z_i), \quad \forall x, z \in (\mathbb{R}^+)^d,$$

where $k$ is an $\gamma$-homogeneous kernel, that is a kernel $k$ on $\mathbb{R}^+$ such that

$$k(cs, ct) = c^\gamma k(s, t), \, \forall c, s, t \in \mathbb{R}^+.$$

As pointed out in [47], this family of kernels is particularly useful on histograms. Exploiting the homogeneous property, the kernel $k$ is rewritten as follows

$$k(s, t) = (st)^{\frac{\gamma}{2}} v(\log s - \log t), \, \forall s, t \in \mathbb{R}^+ \quad \text{with} \quad v(r) = k(e^{r/2}, e^{-r/2}), \, \forall r \in \mathbb{R}.$$

Let $\hat{v}$ be the Fourier transform of $v$. In [47], Lemma 1, they prove that $v$ is a positive definite function, that is equivalent, by the Bochner theorem, to the fact that $\hat{v}(\omega) \geq 0$, for $\omega \in \mathbb{R}$, and $\int \hat{v}(\omega)d\omega \leq V < \infty$. Therefore $k$ satisfies Eq. 6 with $\Omega = \mathbb{R} \times [0, 2\pi]$, a feature map $\psi_0((w, b), s) = \sqrt{V} s^{\frac{\gamma}{2}} \cos(b + w\omega \log s)$, with $(w, b) \in \Omega$ and a probability density $\pi_0$ defined as $\pi_0((w, b)) = \frac{\hat{v}(w)}{V} U(b)$ for all $\omega \in \Omega$, where $U$ is the uniform distribution on $[0, 2\pi]$. Now note that $K$ is expressed by Eq. 6, with the feature map $\psi(\omega, x) = (\psi_0(\omega, x_1), \ldots, \psi_0(\omega, x_d))$ and probability density $\pi(\omega) = \prod_{i=1}^d \pi_0(\omega_i)$. In contrast with the previous examples, [47] suggest to select $\omega_1, \ldots, \omega_m$ by using a deterministic approach, in order to achieve a better accuracy, with respect to the random sampling with respect to $\pi$.

**Infinite one-layer Neural Nets and Group Invariant Kernels [25]**    In this work a generalization of the ReLU activation function for one layer neural networks is considered, that is

$$\psi(\omega, x) = (\omega^\top x)^p \mathbf{1}(\omega^\top x), \quad \forall \omega \in \Omega, x \in X$$

for a given $p \geq 0$, where $\mathbf{1}(\cdot)$ is the step function and $\Omega = X = \mathbb{R}^d$. In the paper is studied the kernel $K$, given by Eq. 6 when the distribution $\pi$ is a zero mean, unit variance Gaussian. The kernel has the following form

$$K(x, z) := \frac{1}{\pi} \|x\|^p \|z\|^p J_p(\theta(x, z)), \quad \forall x, z \in X.$$

where $J_p$ is defined in Eq. 4 of [25] and $\theta(x, z)$ is the angle between the two vectors $x$ and $z$. Examples of $J_p$ are the following

$$J_0(\theta) = \pi - \theta$$
$$J_1(\theta) = \sin \theta + (\pi - \theta) \cos \theta$$
$$J_2(\theta) = 3 \sin \theta \cos \theta + (\pi - \theta)(1 + 2 \cos^2 \theta).$$

Note that when $X$ is a bounded subset of $\mathbb{R}^d$ then Assumption 3 is satisfied. Moreover in [25] it is shown that an infinite one-layer neural network with the ReLU activation function is equivalent to a kernel machine with kernel $K$. The units of a finite one-layer NN are obviously a subset of the infinite one-layer NN. While in the context of Deep Learning the subset is chosen by optimization, in the paper it is proposed to find it by randomization, in particular by sampling the distribution $\pi$.