[Reviews · NeurIPS 2017]

Reviewer 1



Comments: - page 4, 7th line after line 129. "... of features is order sqrt{n}log{n}, ..." -> "... of features is of order sqrt{n}log{n}, ..." - line 134. "This is the first and still one the main results providing a statistical analysis..." -> "This is the first and still one of the main results providing statistical analysis..." - lines 206-207. "It is always satisfied for alpha = 1 ands F = k^2. e.g. considering any random feature satisfying (6). " -> "It is always satisfied for alpha = 1 and F = k^2 e.g. by considering any random features satisfying (6). " ? - line 213, Theorem 3. "...then a number of random features M_n equal to" -> "...then a number of random features M_n is equal to" - line 256. "We f compute the KRR..." -> "We compute the KRR..." - line 276. "... since this require different techniques..." -> "... since this requires different techniques" Conclusion: - The topic of the paper is important, the results are new, the paper is well-written. - Still, the method of random features has various hyperparameters, e.g. the kernel width, the regularization parameter, etc. Moreover, results of Thm. 2 and Thm. 3 depend on parameters r, gamma, etc., which are not known. Thus, we need a model selection approach, as well as its theoretical analysis. - The authors consider problem-specific selection of the sampling distribution. However, it may be even efficient to consider a data-dependent selection of the kernel approximation based on random features. E.g., in the paper "Data-driven Random Fourier Features using Stein Effect" (http://www.cs.cmu.edu/~chunlial/docs/17ijcaiRF.pdf) the authors proposed to estimate parameters of a kernel decomposition in random features based on available input sample. Thus, it can be interesting to consider theoretical properties of this approach using the tools, developed by the authors of the reviewed paper.

Reviewer 2



I was very interested by the result presented here: The main result is that, with a data set of n example, can can acheive an error that decay as 1/sqrt(n) (just as in the Ridge regression method) with sqrt(n)log(n) random projection. I believe this is a very useful guaranty that come as a very good news for the use of random projections and random kitchen sinks, as proposed in NIPS in 2007 Rahimi and B. Recht. Random features for large-scale kernel machines. In NIPS, 2007) Of course, this might be still a lot of random projection, but there are way to obtain them in a fast way by analog means (see for instance the optical device in http://ieeexplore.ieee.org/abstract/document/7472872/ that allows millions of random projections). The gain is indeed interesting from a computational point of view: from O(n^3) and O(n^2) in time and space for standard kernel ridge regression to O(n^2) and O(n√n) with random features Question: I do not see the dimension d playing any role in theorem 1. Section 4 Numerical results could be made clearer. Also, the author could try with data coming from a standard dataset as well.

Reviewer 3



This is in my opinion an excellent paper, a significant theoretical contribution to understanding the role of the well established random feature trick in kernel methods. The authors prove that for a wide range of optimization tasks in machine learning random feature based methods provide algorithms giving results competitive (in terms of accuracy) to standard kernel methods with only \sqrt{n} random features (instead of linear number; this provides scalability). This is according to my knowledge, one of the first result where it is rigorously proven that for downstream applications (such as kernel ridge regression) one can use random feature based kernel methods with relatively small number of random features (the whole point of using the random feature approach is to use significantly fewer random features than the dimensionality of a data). So far most guarantees were of point-wise flavor (there are several papers giving upper bounds on the number of random features needed to approximate the value of the kernel accurately for a given pair of feature vectors x and y but it is not clear at all how these guarantees translate for instance to risk guarantees for downstream applications). The authors however miss one paper with very relevant results that it would be worth to compare with theirs. The paper I talk about is: "Random Fourier Features for Kernel Ridge Regression: Approxiamation Bounds and Statistica Guarantees" (ICML'17). In this paper the authors work on exactly the same problem and derive certain bounds on the number of random features needed, but it is not clear for me how to obtain from these bounds (that rely on certain spectral guarantees derived in that paper) the \sqrt{n} guarantees obtained in this submission. Thus definitely I strongly advice relating to that paper in the final version of this draft. Anyway, the contribution is still very novel. Furthermore, this paper is very nicely written, well-organized and gives an excellent introduction to the problem. What is most important, the theoretical contribution is to my opinion huge. To sum it up, very good work with new theoretical advances in this important field.